# Identification of hot spring Obelisk-like RNA replicons and expanded diversity of the Obelisk superfamily

Syun-ichi Urayama [1,2] ✉, Akihito Fukudome [3,9], Pascal Mutz [4,9], Yosuke Matsushita [5], Yoshihiro Takaki [6], Yosuke Nishimura [6], Sofia Medvedeva [7], Mart Krupovic [7], Eugene V. Koonin [4] & Takuro Nunoura [8]

Recent extensive metatranscriptome mining vastly expanded the range of apparently covalently closed circular (ccc) RNA replicons. A notable family of such replicons is Obelisks, ~1 kilobase (kb) cccRNAs encoding a protein with a unique fold, Oblin-1, and detected in diverse metatranscriptomes. To identify potential cccRNAs in a sequence similarity–independent manner, we adopt the Fragmented and primer-Ligated DsRNA Sequencing (FLDS) method to selectively sequence double-stranded (ds) RNAs, replicative intermediates of RNA replicons. We focus on candidates with predicted extensive intramolecular base-pairing, a hallmark of viroid-like elements. Using FLDS, we explore metatranscriptomes from acidic hot springs in Japan and discover a distinct family of Obelisks apparently associated with thermoacidophilic bacteria (Hot spring Obelisks, HsObs). Despite lacking sequence similarity to known Oblins, HsObs share key features, including ~1 kb genome size, rod-like RNA secondary structure, and the predicted fold of the encoded protein, HsOblin. A comprehensive metatranscriptome search for Oblin-1 and HsOblin homologs expands Obelisk diversity about two-fold, revealing multiple subfamilies sharing the same core fold,. some of which are also predicted to encode additional small proteins with simple alpha-helical folds. These findings highlight Obelisks as widespread and overlooked components of microbial ecosystems, expanding understanding of viroid-like RNA replicon diversity and evolution.

Recently, a broad variety of apparently covalently closed circular (ccc) RNAs have been identified in diverse cells, leading to increased interest in their origins and functions[1–3]. Some of these cccRNAs originate from cellular genomes, whereas others represent RNA replicons including viroids, viroid-like satellite RNAs, and several groups of viruses[4–7]. Viroids are the smallest and simplest known replicators, and the smallest known pathogens as well. Until recently, viroids have been identified only in a few plant species, in some of which they cause

disease. The genomes of viroids are cccRNAs of only 200–400 nucleotides (nt) in size that form extensive rod-shaped or branched secondary structure and encode no proteins[8]. Viroids fully depend on the host cell enzymatic machinery for their replication and are replicated by DNA-dependent RNA polymerases (RNAP) via a version of the rolling circle replication (RCR) mechanism[9]. In one of the recognized families of viroids, *Avsunviroidae*, concatemeric RCR intermediates are processed by hammerhead ribozymes (HHR) present in both polarities

of the viroid RNA[10]. Viroids of the other family, *Pospiviroidae*, lack ribozymes, and the RCR intermediates are processed by host RNases[11].

Apart from the bona fide viroids, several groups of viroid-like cccRNA replicons have been identified. Satellite cccRNAs, sometimes referred to as virusoids, are similar to viroids but depend on plant RNA viruses, being replicated by the viral RNA-dependent RNA polymerase (RdRP) and packaged in the helper virus particles[12]. Retroviroids are viroid-like cccRNA (CarSV) that also contain HHRs, but differ from viroids in that they are transcribed into a DNA form by reverse transcriptases of helper plant pararetroviruses (family *Caulimoviridae*) and can integrate into the plant genomic DNA[13,14]. Retrozymes are small, nonautonomous retrotransposons identified in plant and animal genomes that depend on Ty3-like retrotransposons (family *Metaviridae*) for reverse transcription into a DNA form and integration into the host genome[15,16]. The retrozyme transcripts are circularized by HHRs present in their long terminal repeats.

In addition to these non-coding viroid-like cccRNA replicons, an important human pathogen, hepatitis delta virus (HDV; family *Kolmioviridae*), a satellite of hepatitis B virus (HBV), has a cccRNA genome of about 1.7 kb that encodes a virus nucleocapsid protein (known as Hepatitis Delta Antigen, HDAg) and contains a unique ribozyme in both RNA polarities[17]. Like viroids, the HDV genome is replicated by the host RNAP, whereas HBV provides the virion envelope protein. Until recently, HDV remained the only known viroid-like virus, but in the last few years, related viruses have been discovered in other vertebrates and arthropods[17,18]. Because HDV and its relatives are unrelated to any other known viruses, the International Committee on Taxonomy of Viruses classified this unusual group of viruses into a separate viral realm (the highest rank in the taxonomy of viruses), *Ribozyviria*.

Recent efforts on metatranscriptome mining have transformed our understanding of the diversity and provenance of cccRNA replicons (e.g., ref. 4). Thousands of distinct non-coding, ribozyme-containing cccRNAs in the viroid size range have been identified in metatranscriptomes from diverse environments, many of which are virtually devoid of plant or animal material, implying that these cccRNA replicons are hosted by unicellular eukaryotes and/or prokaryotes. Furthermore, 9 distinct groups of these cccRNAs were found to be targeted by CRISPR systems in genomes of bacteria, strongly suggestive of bacterial hosts[19]. In addition to these viroid-like cccRNAs, metatranscriptome mining led to the identification of several groups of viruses with cccRNA genomes[19]. Many of these are diverse members of *Ribozyviria* encoding distant homologs of HDAg[19,20], whereas others are unrelated to ribozyviruses[19]. A particularly notable group is ambiviruses that possess the largest known cccRNA genomes at nearly 5 kb and encode two proteins, one of which is a distinct RdRP, whereas the other one has no detectable homologs among known proteins. Some of the ambiviruses were previously identified in basidiomycete fungi, but the cccRNA structure of their genomes has not been initially recognized[7,19,21]. Because ambiviruses encode an RdRP, the hallmark of the kingdom *Orthornavirae* within the viral realm *Riboviria*, that shows no specific affinity to any other RdRPs, and given their unique genome organization, ambiviruses are currently classified as phylum *Ambiviricota* within *Orthornavirae*[22].

The most recent major discovery in the expanding domain of cccRNA replicons is Obelisks, cccRNAs of about 1 kb in size encoding a protein without detectable homologs, denoted Oblin-1[23]. Obelisks were originally identified in the human gut metatranscriptomes but subsequently found in a broad variety of environments and appear to be particularly abundant in marine ecosystems[24]. For one of the Obelisks, the bacterium *Streptococcus sanguinis* has been definitively identified as the host[23].

In this study, we use "covalently closed circular RNAs (cccRNAs)" as an operational term. These RNA assemblies exhibit terminal redundancy and highly stable, rod-like secondary structures, features commonly associated with circular viroid-like replicons. However, consistent circular assemblies alone do not establish that these RNAs exist as covalently closed circles. Current metatranscriptomic approaches, including FLDS, cannot distinguish cccRNAs from concatemeric repeat assemblies and alternative RNA topologies that could be present in metatranscriptomes. Thus, the ccc structure of these molecules has not been experimentally demonstrated and remains a hypothesis, even if one that is best compatible with the available data.

In this work, we explored dsRNA metatranscriptomes constructed from acidic geothermal spring microbiomes by using Fragmented and primer-Ligated DsRNA Sequencing (FLDS)[25,26], a method for the selective sequencing of dsRNA, a typical replicative intermediate of RNA replicons[27–29]. Among the metatranscriptomes, some of which harbor linear genomes of the realm *Riboviria*, such as HsRV and HsPV[30], indicating that RNA replicons can persist in such extreme ecosystems. In addition, the exploration of cccRNAs with predicted extensive intramolecular base-pairing that initiate dsRNA, a hallmark of viroid-like elements, from the dsRNA metatranscriptome offers advantages for structure-based detection of dsRNA rather than sequence similarity. CccRNAs are typically even smaller than RNA viruses and are often undetectable by homology-based searches. Here, we identified a distinct family of Obelisks from dsRNA metatranscriptomes from geothermal acidic hot springs in Japan. A comprehensive search for Oblin-1 homologs substantially expanded the diversity of the Obelisk superfamily of cccRNA replicons, notably revealing their presence in high-temperature environments beyond previously studied gastrointestinal and aquatic ecosystems.

## Results

### Validation of the cccRNA detection workflow on a dsRNA-seq dataset containing a known viroid

As a proof-of-concept demonstration of our FLDS-based workflow, we first analyzed dsRNA-seq data from plant leaves containing a known viroid, Chrysanthemum stunt viroid (CSVd, species *Pospiviroid impedichrysanthemi*). The dsRNA-seq data can include reads originating from cccRNA replicons because RNA-dependent RNA replication, regardless of the specific mechanism involved, requires transient formation of a template–substrate dsRNA duplex[9,31,32] and possibly from matured cccRNAs with extensive intramolecular base-pairing[27]. We established a data processing workflow to identify candidate cccRNA replicons from FLDS data in a sequence similarity-independent manner (Fig. 1). Of the 6311 contigs generated by de novo assembly of clean FLDS reads using SPAdes (metaplasmid mode), 7 contigs were predicted by the ccfind program (see Code availability) to be cccRNAs, based on terminal redundancy. Based on the BLASTX and BLASTN search results, three of these sequences were discarded because they showed similarity meeting the BLAST threshold ($E$-value ≤ $1 \times 10^{-05}$) to known plant and human sequences. Two additional sequences were removed because of their low coverage (<10 reads per nucleotide on average). The two remaining sequences with more than 10 reads per nucleotide on average were predicted to form an extensive secondary structure (>60% paired bases) (see "Methods"). Using read mapping data, we assessed the evenness of coverage for each contig by calculating two metrics: the normalized coefficient of variation of read depth and the normalized entropy of read start positions. As an empirical criterion based on previous FLDS analyses, contigs derived from bona fide dsRNA molecules tend to show relatively uniform read coverage[26]. Contigs with uneven mapping (normalized CV of 0.6 or higher or normalized entropy of 0.7 or lower) were excluded, and only those with uniform mapping were retained for downstream analyses (Table S1; see "Methods"). The BLASTN search showed that these two sequences were variants of CSVd, with more than 95% nucleotide identity to the CSVd reference sequence (accession: X16408) (Fig. 1). These results indicate that the combination of FLDS and the developed workflow with the appropriate filtering enables the structure-dependent identification of cccRNA replicons.

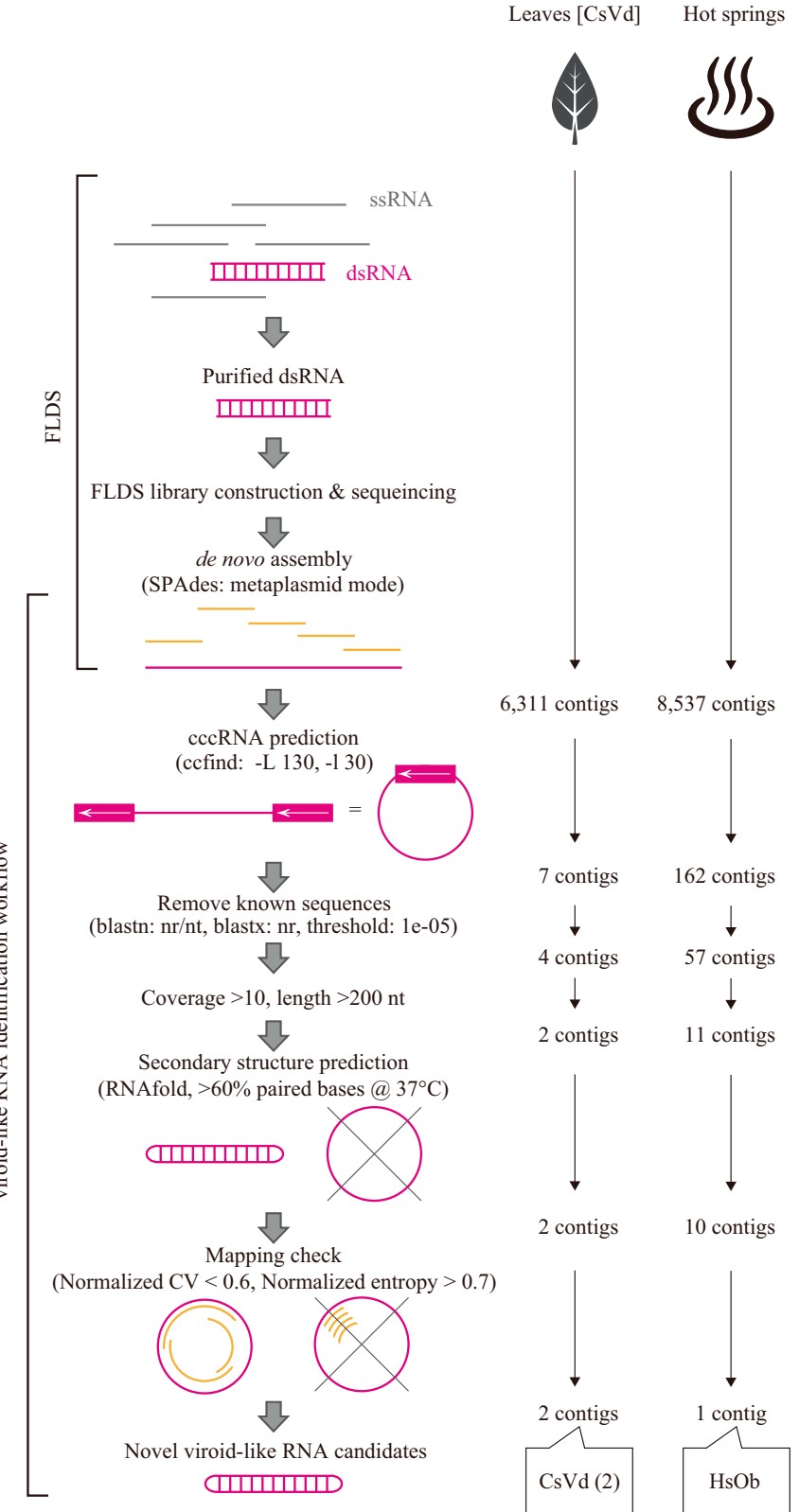

**Fig. 1 | Schematic workflow of cccRNA replicon surveillance.** DsRNAs obtained from samples were sequenced using the FLDS method[30]. The contigs assembled through de novo assembly were used for circular RNA prediction. After removing sequences with low coverage, those that were too short, and duplicated sequences, the predicted secondary structure and mapping patterns were also checked. Details of this workflow are described in the "Methods" section.

**Table 1 | Characteristics of hot spring water samples**

| Code | Geographical coordination | Area | Temp (°C) | pH | DO (mg/L) | H₂S (mM) | Sampling Date | HsOb reads/total reads[b] |
|---|---|---|---|---|---|---|---|---|
| H4 | 31°54′07.5″N 130°50′06.2″E* | Hayashida | 68.8 | 3.2 | 2.1 | 1.3 | 10-Mar-2017 | |
| H5 | 31°54′07.5″N 130°50′06.2″E | Hayashida | 69.7 | 3.1 | 2.0 | 1.8 | 10-Mar-2017 | 56/1,734,416 |
| T1 | 31°54′37.7″N 130°49′00.6″E | Tearai | 92.1 | 2.9 | - | 0.0 | 09-Mar-2017 | |
| T2 | | Tearai | 95.9 | 2.1 | - | 0.0 | 09-Mar-2017 | |
| T3 | | Tearai | 94.4 | 2.4 | 0.0 | 0.0 | 09-Mar-2017 | |
| T4 | | Tearai | 92.8 | 2.7 | 0.0 | 0.0 | 09-Mar-2017 | |
| Y66 | 31°55′03.8″N 130°48′40.4″E | Yunoike | 68.7 | 2.7 | 2.1 | 0.0 | 10-Mar-2017 | |
| Y80 | | Yunoike | 75–86[a] | 2.5 | 1.5 | 0.0 | 10-Mar-2017 | |
| Y86 | | Yunoike | 86.5 | 2.5 | 0.0 | 0.0 | 10-Mar-2017 | |
| Oi | 32°44′25.3″N 130°15′48.4″E | Unzen | 79.3 | 2.2 | 0.0 | 0.4 | 18-Nov-2015 | 408/316,160 |
| Ob | 32°43′33.0″N 130°12′24.7″E | Obama | 72.8 | 7.9 | 0.0 | 0.0 | 17-Nov-2015 | |

*Data other than read counts are reproduced from Urayama et al.[30].
[a]There were temperature gradients in the pool site: surface layer 75.0 °C; bottom layer 81.6 °C, 80.3 °C, 85.9 °C; middle layer 81.0 °C.
[b]Mapping: Length fraction = 0.9, Similarity fraction = 0.9.

## cccRNAs from acidic geothermal spring dsRNA metatranscriptomes

Using the validated workflow described above, we analyzed dsRNA metatranscriptomes from 11 high-temperature acidic geothermal hot spring samples (Fig. 1, Table 1). Using the ccfind program, analysis of 8,537 contigs obtained from the de novo assembly of FLDS metatranscriptomes for each sample yielded 162 candidate cccRNAs. Among these sequences, 105 were discarded because they showed similarity meeting the $E$-value threshold ($E$-value $\leq 1 \times 10^{-05}$) to known non-cccRNA sequences in NCBI nr/nt or nr (protein) databases using BLASTN and BLASTX, respectively. After discarding short sequences (<200 nt) and those with low average sequencing coverage (<10), 11 candidate cccRNAs remained (Table S1). Of these, 10 were predicted to form a viroid-like secondary structure using RNAfold (>60% paired bases). Nine of these sequences were excluded from further consideration due to uneven read mapping, as determined by their normalized CV and entropy values (Table S1). Finally, a candidate sequence, Hot spring Obelisk (HsOb) (see below), was distinct.

## Obelisk-like elements from hot spring dsRNA metatranscriptomes

The single relatively large cccRNA and a related sequence were detected in the dsRNA metatranscriptome from two geographically distinct sampling stations, Oi and H5 (see below, Table 1). The Oi and H5 hot springs are characterized by high temperatures (69.7 °C and 79.3 °C) and low pH (3.1 and 2.2), respectively. The rRNA of potential host microbial communities from Oi and H5 were both dominated by thermophilic bacteria of the family *Hydrogenobaculaceae* in the phylum *Aquificota* (49.8 and 98.9%, respectively)[30].

The cccRNA sequence obtained from the acidic hot spring Oi (79.3 °C, pH 2.2) consisted of 867 nucleotides with a GC content of 49.8% and showed no detectable similarity under the BLASTN threshold to the sequences in the NCBI nucleotide (nr/nt) database. The secondary structure of the HsOb sequence predicted using RNAfold at the default temperature (37 °C) resembled a long, unbranched rod with extensive intramolecular base pairing (68%) (Fig. 2A). At the physiologically-relevant temperature of 80 °C, the predicted long rod-like structure was retained, although with a considerable decrease in base pairing. Although base pairing decreased substantially at 80 °C, the rod-like structure was retained in these conditions. Because secondary-structure prediction at high temperatures has limited accuracy, these results are best interpreted qualitatively, suggesting partial local melting rather than a major change in topology. To further characterize the cccRNA, self-cleaving ribozymes were predicted using RNA sequence and secondary structure covariance models using Infernal[33]; however, no known self-cleaving ribozymes were identified.

The cccRNA encompasses a single long ORF encoding a putative protein of 213 amino acids, which occupies approximately three-quarters of the nucleotide sequence (Fig. 2B). Although the amino acid sequence of the putative ORF product did not show detectable similarity under the BLASTP threshold ($E$-value $\leq 1 \times 10^{-05}$) to the NCBI nr protein sequences, structural modeling using AlphaFold2 (AF2[34]) and AlphaFold3 (AF3[35]) yielded confident models of the HsOb-encoded protein (AF2, pLDDT = 84.5, ptm = 0.62; AF3, pLDDT = 70.4, ptm = 0.59; Figs. 2C and S1A). Including RNA such as poly-U RNA or a terminal hairpin RNA (nt 324–337) in AF3 predictions improved confidence in the C-terminal region. The average pLDDT scores for residues 152–213 increased from 49.97 in the absence of RNA to 68.19–73.39 upon RNA inclusion, accompanied by an increase in the overall pTM scores from 0.59 to 0.78–0.80 and high ipTM scores (0.88–0.90) (Fig. S1B, C). Consistent with these changes, the C-terminal region exhibited high predicted aligned error (pAE >25 Å) without RNA, whereas inclusion of RNA reduced pAE values to medium or low expected error (<5 Å) across many positions (Fig. S1C; see color bar). We also tested RNA addition with Obelisk-alpha Oblin-1 AF3 prediction, but it failed to generate a confident RNA-protein complex model and showed no consistent improvement in the C-terminal region, either with a poly-U 14-nt RNA or a terminal hairpin RNA (moderate pAE only in ssRNA, Fig. S1D). Comparison of the AF2 models of the HsOb-encoded protein and Oblin-1 revealed notable structural similarity (Root Mean Square Deviation of 3.817 Å as calculated using the MatchMaker tool of ChimeraX; see "Methods") despite the low sequence similarity (12.5% identity) (Fig. 2C). The shared predicted structural features included an N-terminal globular domain with α-helical bundles bookended by a two-stranded β-sheet, and an apparently flexible C-terminal region containing multiple positively-charged residues in the region around the position equivalent to the conserved domain-A in original Oblin-1, although there was no detectable amino acid sequence motif "GYxDxG" in this region of the ORF product (Figs. 2C, D and S4). These observations indicate that the cccRNAs identified here encode a homolog of Oblin-1 with a high degree of structural conservation between the N-terminal alpha-helical bundle domains. Based on the shared genomic organization and the similarity of the encoded protein predicted structures, we concluded that the cccRNAs from the Oi hot spring represent a distinct group of Obelisks[23] that we named Hot spring Obelisk-like RNA strain Oi (HsOb-Oi) (accession: LC895899), denoting the encoded protein HsOblin-1-Oi.

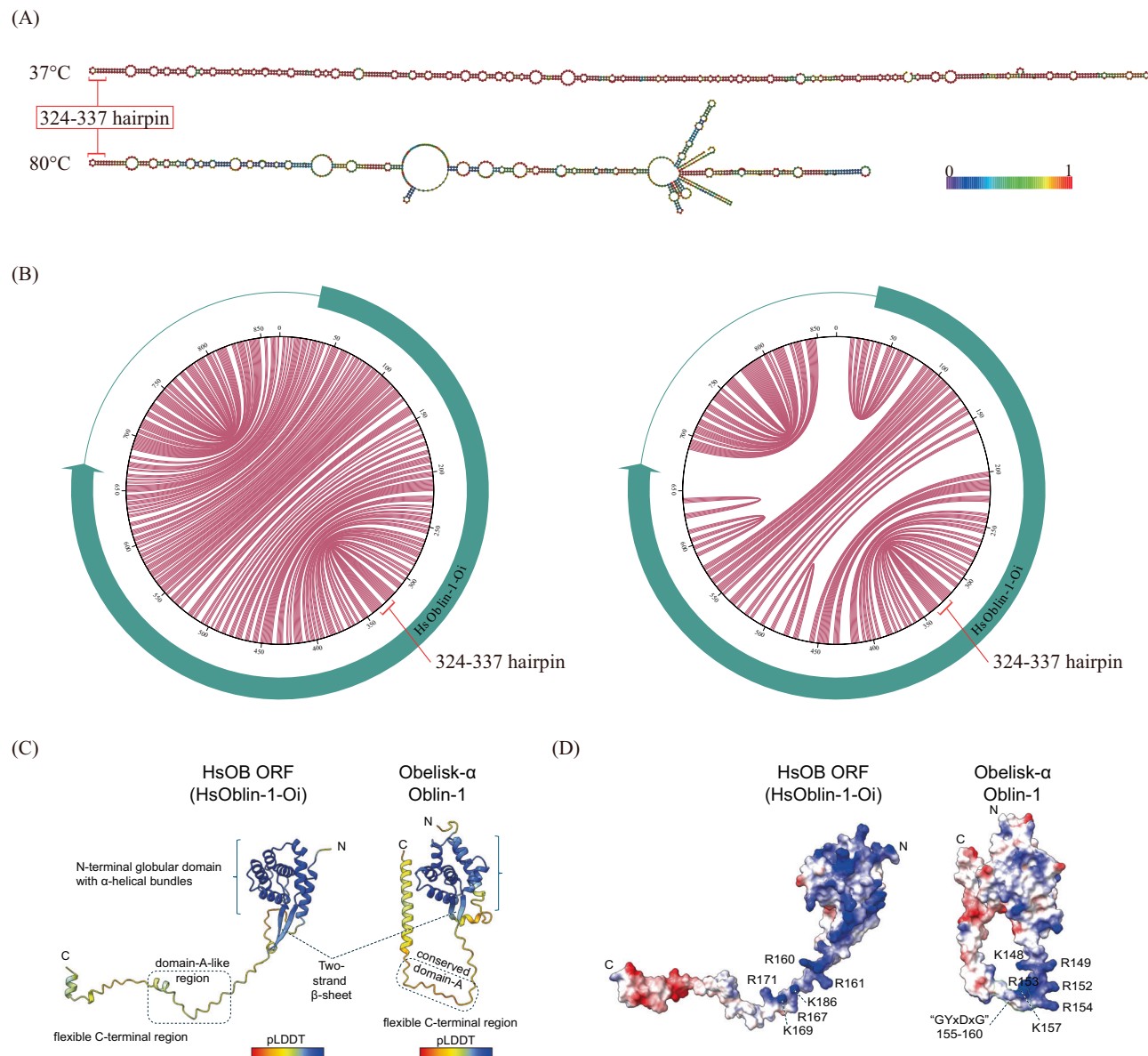

**Fig. 2 | Characteristics of HsOb-Oi. A** Schematic of the sense secondary structure derived from HsOb at 37 °C and 80 °C predicted by RNAfold. The ruler color indicates base-pairing probabilities. **B** Jupiter plot of HsOb at 37 °C and 80 °C and predicted open reading frame. **C** AlphaFold2 structural models of HsOblin-1-Oi and Oblin-1 from the Obelisk-α. Ribbons are colored by pLDDT scores, and the color key

is shown. **D** Electrostatic surface potential of the HsOblin-1-Oi and Oblin-1 protein models. Blue and red represent positive and negative changes, respectively. Positively charged residues around the domain-A region are labeled. The GYxDxG motif in Oblin-1 is highlighted by a green border line.

In addition, we performed a FoldSeek search of protein structure databases using the confidently modeled N-terminal globular domain of the ORF product (aa51–148) as the query, but no detectable structural similarity was found in any of the databases (the best $E$ value was 0.31). Notably, however, these databases did not include the predicted structure of Oblin-1.

We then explored the sequence diversity of the HsOblin-1-Oi by searching the FLDS metatranscriptomes obtained from other hot springs. The amino acid sequence of the HsOblin-1-Oi was used as a query to search against the SPAdes-assembled contigs using BLASTX. This search yielded an additional HsOblin-1 sequence ($E$-value = $1 \times 10^{-108}$; 71.7% pairwise amino acid sequence identity) encoded by a contig from sample H5. The H5 contig, denoted HsOb-H5, was incomplete (we could not recover a circular sequence), but encoded a full-length HsOblin-1-Oi homolog, which we denoted HsOblin-1-H5 (accession: LC895898).

In an attempt to assign the host for HsObs, we searched their sequences against 720 CRISPR spacer sequences obtained by metagenomic DNA sequencing (DRR898023-DRR898026) of the hot spring samples[30] as well as 40,704 *Sulfolobales* spacers from the Beppu hot springs[36]. However, no spacers matching the HsOb genomes were identified.

## Major expansion of the Obelisk and Oblin-1 diversity and prediction of Obelisk hosts

To identify additional Obelisks and Oblins related to HsOblin-1 and the prototype Oblin-1[23], we mined ~8.9 million putative cccRNAs from more than 5000 metatranscriptomes from a recent study[19] and additional ~4 million putative cccRNAs from about 2000 assembled metatranscriptomes which more recently became available at IMG/MER[37], for homologs of Oblin-1 and HsOblin-1-like proteins using an

iterative search procedure (see "Methods"). This search revealed 6867 deduplicated Obelisk sequences encoding 5009 deduplicated Oblin-1 and HsOblin-1 homologs. The retrieved homologous sequences were supplemented with about 1700 previously identified Oblin-1 centroid proteins[23], the FLDS identified HsOblin-1-Oi and -H5, and 5 (1 full-length and 4 partial sequences) HsOblin-1 proteins identified by BLASTX in Yellowstone hot spring sequencing projects (see "Methods"), bringing the total number of Oblin-1 homologs to 6443 deduplicated sequences. The Obelisk set was clustered at 80% average nucleotide identity (ANI) to follow the previously proposed nomenclature[23], resulting in 1774 clusters spanning known and newly identified Obelisks and 2060 clusters without known Obelisks, expanding the Obelisk diversity about two-fold at this level (see "Methods").

This protein set was clustered by sequence similarity, yielding 111 clusters (see "Methods"). The relationships among these clusters remained uncertain because of the low sequence similarity approaching the limit of the HMM comparison resolution. The majority of large clusters (10 or more members) contained known Oblin-1 proteins, in addition to those discovered in this work, but 3 clusters consisted of new sequences only (Fig. 3, Clades C1-3), including the HsOblin-1-like cluster, C3, and one cluster contained only 4 known centroids (O18). The C3 cluster containing HsOblins and Yellowstone HsOblin-1 consisted of 44 sequences. For each cluster, a phylogenetic tree was built, midpoint rooted, and grafted onto the similarity dendrogram (Fig. 3A). Structure prediction from a representative set of diverse sequences across each clade supported the presence of an Oblin1-like fold in all these proteins, although the two-stranded β-sheet was not always predicted with high confidence. Clustering 40 putative Oblin-1 sequences from a recent study[24] with the present set at 50% sequence identity indicated that 19 of the 40 mapped to clade O13, 6 to O18, and 15 represented partial or diverged sequences that could not be mapped. Notably, for clade O18, a third beta-strand was predicted, interacting with the classical beta sheet clasp (Fig. 3B). Proteins in this clade are most closely related to a distinct set of several Oblin-1-like proteins reported by Zheludev et al.[23] which also show a third beta-strand in their structure prediction (O16 and O17 in Fig. 3A), therefore substantially expanding this Oblin-1-like protein clade. Although structure comparison with Dali can yield low z-scores at the border of similarity detection even when comparing only the N-terminal globular domain with α-helical bundles bookended by the two-stranded β-sheet, the topology of the alpha-helical globular domain is conserved across the different newly discovered Oblin-1 cladess (Fig. S2). The majority of the cccRNAs encoding Oblin-1 homologs were within the 800–1400 nt size range. The Oblin1-like ORF is typically between 400–1000 nt long, resulting in a cccRNA coverage (length ORF/length monomeric cccRNA) with the Oblin-1 ORF between 0.5–0.9 for the majority of sequences (Fig. 3C).

The phylogenetic tree indicated clustering of the sequences originating from diverse ecosystems (Figs. 4 and S3). Aquatic and terrestrial environments dominate the respective clades and, for the C3 clade, sequences found in hot spring ecosystems cluster together on the tree. Three other clades included a few Oblins from hot springs, all from bioprojects related to Yellowstone National Park hot springs (C2, O11, and O13 with 2, 2, and 1 Oblin, respectively). Otherwise, Oblins from aquatic and terrestrial environments dominated clusters C1, C2, O18, and O13, a clade including known Oblin-1 proteins of the phi group but otherwise substantially expanded in this study. In an attempt to gain additional information on putative bacterial or archaeal hosts, we performed a CRISPR spacer search (see "Methods") for all newly identified Obelisks. Searching a local, well-curated spacer database, only one Obelisk matched 29 nucleotides of a single 32 nt spacer from an RNA-targeting CRISPR system (Cas-III-B) of the bacterium, *Marinomonas mediterranea* (order *Gammaproteobacteria*). In addition, we searched the high-quality spacer dataset from JGI (https://spacers.jgi.doe.gov/, ref: https://www.biorxiv.org/content/10.1101/2025.06.12.659409v1) with newly identified and known Obelisks using MMSEQS2[38] (see "Methods" for detailed parameters). This search retrieved 63 unique spacers with reliable matches to 278 Obelisks, of which the majority were known or closely related to known ones (Fig. S11). All these Obelisk-matching spacers were at least 30 nt long, and 85% included no or a single mismatch. Most spacers were not assigned to a specific CRISPR type, but 9 were assigned to type III CRISPR systems that are capable of integrating RNA-targeting spacers into CRISPR arrays via reverse transcription catalyzed by a reverse transcriptase (RT) embedded in the adaptation module[39]. The 63 unique Obelisk-matching spacers are associated with several hosts, with the majority (88%) coming from species of the bacterial families *Lachnospiraceae* and *Selenomonadaceae* (both phylum *Bacillota*), known to colonize the gastrointestinal tract of humans and ruminants (see Figs. S11 and 12 and supplementary material for full spacer overview including a complete species list). Hence, it is likely that two of the most associated species, RUG12372 sp900321035 (family *Selenomonadaceae*) and UBA1066 sp902776305 (*Lachnospiraceae*), represent true hosts for Obelisks. In addition, 404 high-quality annotated reference genomes (from complete prokaryotic genomes obtained from the NCBI GenBank in November 2023 (prok2311)) of species from the putative host families were inspected for the presence of RTs within the CRISPR loci. Although none of the putative host species associated with the matching spacers was present among the reference genomes, we found three related species in *Bacteroidaceae* and one species in *Lachnospiraceae* carrying RT-encoding type III CRISPR loci, indicating that, in principle, at least, integration of RNA-targeting spacers into CRISPR arrays can occur in bacteria of these families (see supplementary material at zenodo "spacer_search"). Of note, several Obelisks were targeted by multiple spacers from bacteria of these families (Fig. S13), increasing the confidence that these are true hosts of Obelisks.

Because HsOblin-1-Oi/H5 amino acid sequences lack the "GYxDxG" motif, which is the signature of domain-A in the original Oblin-1 proteins[23], we searched for sequence motifs conserved across the expanded cluster of 44 HsOblin-1-like proteins (C3) in the C-terminal region corresponding to the domain-A of Oblin-1s. HsOblin-1-like proteins encompassed three larger conserved regions that in the protein structure were located in positions corresponding to those of the conserved regions of known Oblin-1 proteins, two within the globular alpha-helical domain and one within the flexible C-terminal portion of the protein (Fig. 5). Similarly, no conserved GYxDxG motif could be detected for clades C1 and C2 as well as for O18, underpinning its status of a new Oblin-1 clade. Similar to HsOblin-1-like proteins, proteins from clades C1, C2, and O18 contain a stretch of variable length of positively charged residues. O18 contains, in addition, a highly conserved serine-glutamine pair, C1 contains two conserved serines (S(A/P/-)S), and C2 contains fewer positively charged residues but three tyrosines and two serines ((R/P)YSSY(C/T)KGYG) (Fig. S4). By contrast, Oblin-1 proteins from clades O16 and O17 retain the conserved GYxDxG motif.

The secondary structures of Obelisk RNAs from hot springs and other newly identified clades were predicted using RNAfold[40]. As expected, given the previously predicted rod-like structure of Obelisks as well as many other cccRNAs replicons, stable structures were predicted, the average minimum free energy for the majority of the clades being below −300 kcal/mole (Fig. S7A, B). In line with the predicted high stability of the Obelisk RNAs, the average fraction of paired bases was between 68–72% except for clade O9 (64% paired bases, sense strand) (Fig. S5C, D). As previously observed[23], the majority (82% on average) of bases of the Oblin-1 ORF involved in base pairing are "self-complementary," that is, paired to other bases within the same ORF (Fig. S15).

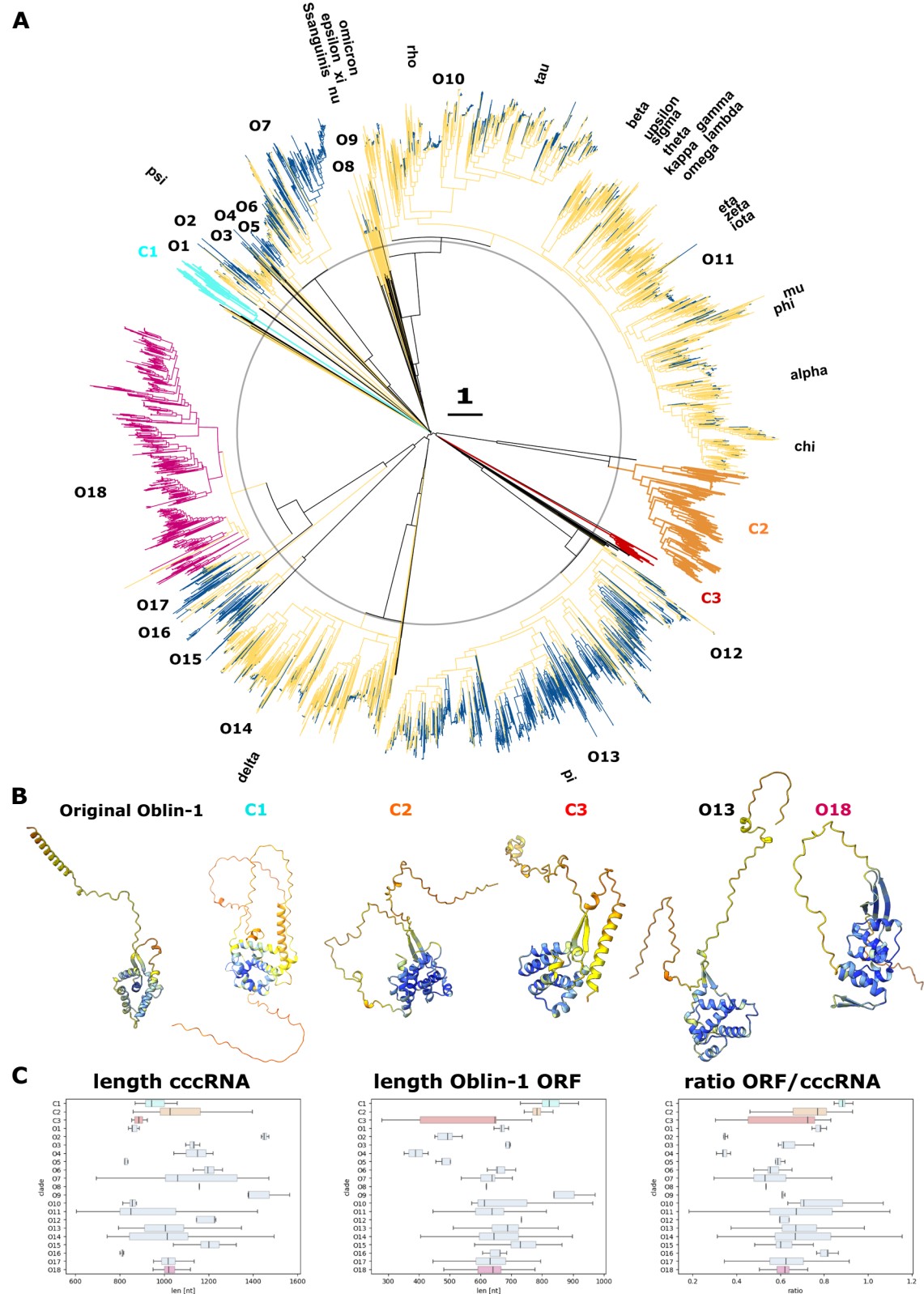

**Figure A** (circular phylogenetic tree with clades labeled C1, C2, C3, O1–O18, and Greek-letter clades: psi, omicron, epsilon, xi, nu, rho, tau, beta, upsilon, sigma, theta, gamma, kappa, omega, lambda, eta, zeta, iota, mu, phi, alpha, chi, pi, delta)

**Figure B** — Original Oblin-1, C1, C2, C3, O13, O18 (protein structures)

**Figure C**
- length cccRNA
- length Oblin-1 ORF
- ratio ORF/cccRNA

We searched all Obelisks for the presence of ribozymes using Infernal[33]. Hammerhead ribozymes (HHR) were predicted in more than 800 Obelisks, mapping to more than 600 leaves in the Oblin-1 tree. The majority of these were HHR3, located in the strand complementary to the Oblin-1 encoding strand. Only a few HHRs other than HHR3 or ribozymes in the sense strand were predicted (Fig. S6). Mapping ribozymes in relation to the Oblin-1 ORF indicated that the majority (70%) is on the antisense strand, non-overlapping with the Oblin-1 ORF, and has a median distance of 51 nucleotides to the 5' end (start codon) of the Oblin-1 ORF. Other positions, including completely nested ribozymes inside the Oblin-1 ORF, were detected at lower frequencies (e.g., nested ribozymes in 22% of cases; Fig. S7). The HHR3 were predicted mainly in Obelisks of the Oblin-1 clade O18 and associated clades O16 and O17, which is compatible with the original observations

**Fig. 3 | Overview of known and newly identified Oblin-1 proteins. A** Inner part (till black circle): dendrogram based on pairwise hhsearch scores (including artificial low score in case no relationship was detected by hhsearch) of Oblin-1 clusters. Outer part: Midpoint routed FastTree phylogenetic trees based on alignment of Oblin-1 sequence, which could be aligned confidently in an iterative procedure with hhalign. Yellow clades indicate those related to known Oblin-1 proteins (centroids of ref. [23]) with dark blue terminal branches indicating sequences found in this study. Three larger clades (C1, C2, and C3), including HsOblin-1-like (red, C3), don't harbor any known centroid Oblin-1 proteins. O18 contains only four known Oblin-1 sequences. **B** Representative structure predictions per selected clades of known

and novel Oblin-1-like proteins. **C** cccRNA length, Oblin-1 ORF length, and their ratio (from left to right) per clade (for clades with 10 or more members). Boxplots show borders of first quartile, median, and third quartile as a box, whiskers show the last datapoint within 1.5× the interquartile range. Sequences (both newly discovered and known) (n=) per clade: C1: $n = 82$; C2: $n = 459$; C3: $n = 60$; O1: $n = 65$; O2: $n = 42$; O3: $n = 106$; O4: $n = 18$; O5: $n = 20$; O6: $n = 175$; O7: $n = 568$; O8: $n = 20$; O9: $n = 16$; O10: $n = 137$; O11: $n = 2560$; O12: $n = 14$; O13: $n = 1611$; O14: $n = 783$; O15: $n = 151$; O16: $n = 19$; O17: $n = 185$; O18: $n = 1291$. Source data are provided as a Source data file (Source_data_fig.zip).

of Zheludev et al.[23]. Also, the majority of ribozymes discovered here are found in comparable positions as reported by Zheludev et al.: a non-overlapping antisense HHR3 ribozyme in close proximity to the 5′ end of the Oblin-1 ORF and, if detected, a nested sense HHR3 ribozyme at the 3′ end of the Oblin-1 ORF.

### Diversity of non-Oblin-1 proteins encoded by Obelisks

The mean HsObs cccRNA size was 892 nt (range of 853–995 nt), whereas the mean HsOblin-like ORF length, including the complete one from the Yellowstone hot spring metatranscriptome, was 599 nt (range of 279–765 nt). ORFs shorter than 350 nt encode partial Oblins covering the C-terminus, indicating the possibility of alternative start codon use. The mean cccRNA coverage by the HsOblin-1-like ORF was 67% (range of 30–82%), leaving room for potential additional proteins. Comparable observations were made for cccRNAs from other Oblin-1 clades. To investigate the coding potential of Obelisks, amino acid sequences encoded by additional ORFs of the newly discovered and original Obelisks were extracted (see "Methods"). The 27,926 dereplicated putative non-Oblin-1 proteins were clustered by similarity (40% identity, 70% coverage), resulting in 18,609 clusters including 14,381 singletons. Of the 4228 non-singleton clusters (2–48 members), 545 contained five or more members. The protein sequences from the non-singleton non-Ooblin-1 clusters were aligned, converted into HMM profiles, and compared to each other using hhsearch. Cluster representatives were modeled using AlphaFold3 and compared to each other using foldseek. Structure comparison did not result in meaningful clusters due to the low prediction confidence for the majority of the sequences (Fig. S8) and the simplicity of the folds (mostly, small all helical domains with or without disordered regions). Therefore, only profile comparison was considered to define homologous protein groups. Members of these groups were mapped onto the Oblin-1 phylogeny to identify non-Oblin-1 proteins associated with the respective Oblin-1 proteins.

None of the putative non-Oblin-1 proteins encoded by HsObelisks or other new clades showed detectable sequence similarity to the putative second protein encoded by the prototype Obelisks, Oblin-2. Furthermore, Oblin-2 proteins identified by PSI-BLAST using the prototype Oblin-2 as the query showed a patchy representation across the Oblin-1 tree and appeared mainly in subclades of Oblin-1 clades O11 and O14 (Fig. S9). In contrast, several other clusters of putative non-Oblin-1 proteins were strongly associated with distinct Oblin-1 branches. In particular, 10 non-Oblin-1 protein clusters mapped to Oblin-1 clades in which 70% or more of the Obelisks encode the respective non-Oblin-1 protein (Fig. S9 groups 0–9). Further, the two largest non-oblin-1 structure clusters mapped to Oblin-1 clades with a lower coverage (Fig. S9, lc1 and lc2). Inspection of the predicted structures of putative non-Oblin-1 proteins and the genome organizations of the respective Obelisks highlights 6 non-Oblin-1 protein clusters with confidently predicted structures and no overlaps with the Oblin-1 ORF, all associated with Oblin-1 clade O18 (Figs. S9 and S10 groups 0, 2, 6, 7, and lc2). In 5 of the 6 cases, the non-Oblin-1 ORF is encoded on the reverse strand compared to the Oblin-1 ORF (Fig. S10). Notably, in these 6 cases, the Oblin-1 and non-Oblin-1 ORFs are largely not base-paired in the cccRNA

secondary structure prediction (both sense and antisense), demonstrating their capacity to evolve independently (Figs. S9B and S10).

Taken together, the above observations strongly suggest that at least the latter 6 clusters of non-Oblin-1 ORFs associated with O18 Oblin-1 clade actually encode proteins. Predicted folds of these non-Oblin-1 proteins are relatively simple alpha-helical structures, including a beta-hairpin in three groups (Fig. S10, non-Oblin-1 groups 2, 7, and 9). Profile- and structure-based comparisons with various databases did not show confident hits for any non-Oblin-1 protein group. Given that Oblin-2 proteins were reported to contain a leucin zipper motif[23], coiled-coil regions were predicted for 12,774 representative non-Oblin-1 proteins (CoCoNat[41]); 1338 non-Oblin-1 proteins with putative coiled-coil regions of at least 21 amino acids were inspected for $i + 7$ leucine stretches characteristic of leucine zippers (see "Methods" for details). Requiring at least 3 heptad repeats [21 amino acids] in which leucine was found at the "a" position of the heptad and either leucine or isoleucine at the "d" position in 50% of the cases and at least three matches either at the "a" of the "d" position indicate that 244 non-Oblin-1 proteins might form leucine zippers, with only 2 of them identified as Oblin-2.

The actual existence of non-Oblin-1 Obelisk-encoded proteins apart from these 6 clusters, including Oblin-2, remains an open question.

## Discussion

Recent metatranscriptome mining has greatly expanded the known diversity of viroid-like cccRNAs[19,23,24]. To further explore these putative replicons in high-temperature environments, we applied FLDS to samples from the acidic geothermal hot springs in Japan. The HsOb was identified in a microbial community dominated by thermo-acidophilic bacteria taken from a spring water (79.3 °C), in which ribovirus RNA-dependent RNA polymerase (RdRP) sequences were also detected[30]. The HsOb shares the same genomic organization as Obelisks. The HsOblin-1 AlphaFold models reveal conservation of key structural features among Oblin-1 proteins, including N-terminal α-helical bundles followed by a positively charged C-terminal region. The improvement of HsOblin-1-Oi AF3 model upon inclusion of RNA, particularly at the C-terminal region, is consistent with a hypothesis that HsOblin-1-Oi could directly interact with RNA, potentially, via the conserved positively charged region, as previously discussed for Oblin-1 proteins by Zheludev at al.[23]. However, the variability of predicted protein-RNA interfaces with different RNAs, together with the lack of clear improvement in Obelisk-α Oblin-1 AF3 models from RNA addition, call for caution in interpretation. The predicted HsOblin-1-Oi-RNA interaction models remain to be evaluated experimentally in the context of HsOb/Obelisk replication. Based on our results, the replication of HsOb is likely similar to that of previously reported Obelisks, although the involvement of RdRPs cannot be completely ruled out.

Previous metatranscriptome mining identified several groups of small, apparently non-coding cccRNAs in bacteria-dominated hot springs from Yellowstone, with the temperature of approximately 60 °C[4]. For one of these non-coding cccRNAs, CRISPR targeting was demonstrated, strongly suggesting that this element replicates in a

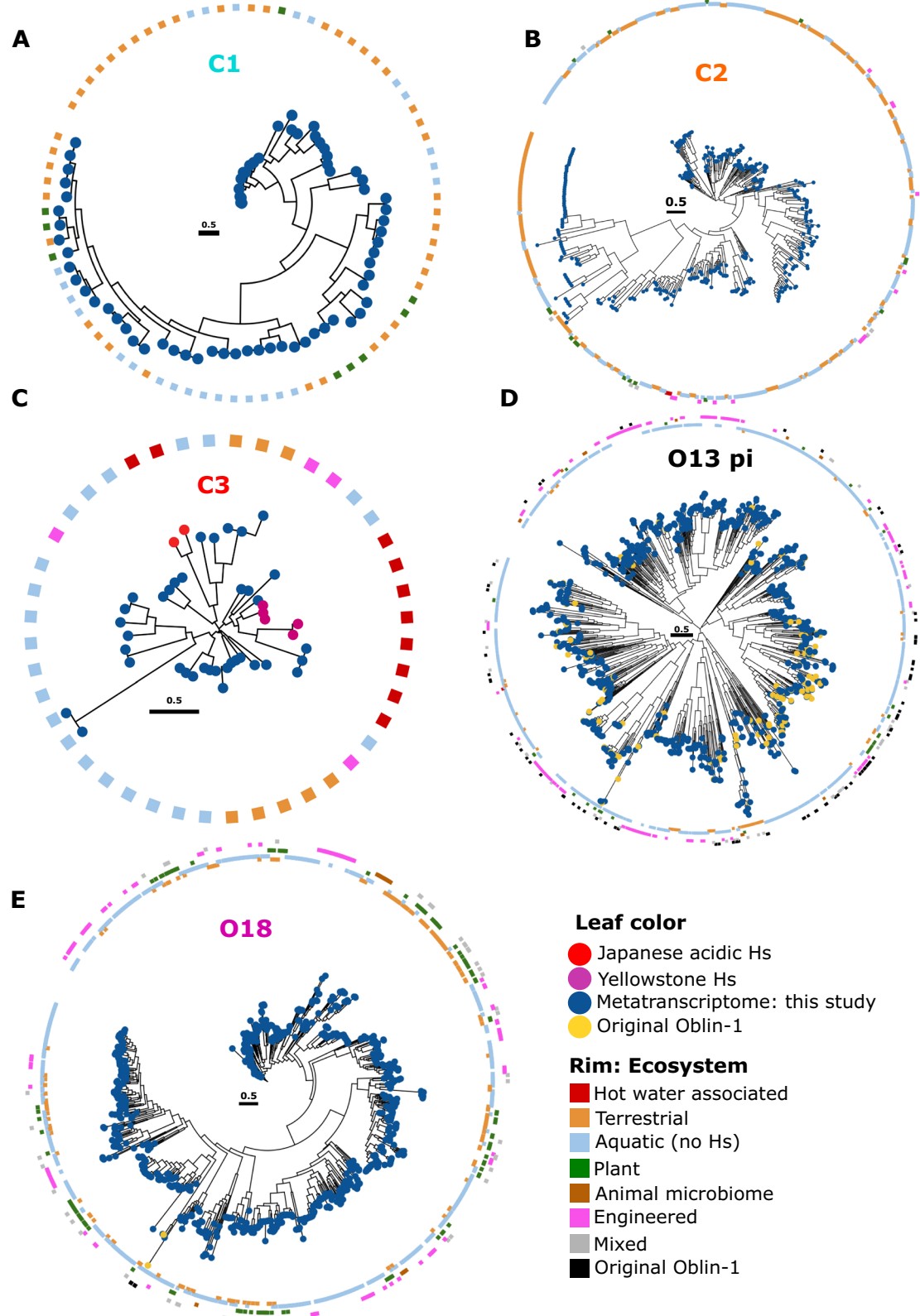

**Fig. 4 | Ecosystems positive for Oblin-1 proteins from different subfamilies.**
**A–E** Unrooted phylogenetic Oblin-1 trees for selected clades are shown (clade nomenclature as in Fig. 3). Outer rim indicates the respective overarching ecosystem of the metatranscriptome study. Leaf color indicates Oblin-1 proteins discovered in this study (blue) or centroids from Zheludev et al.[23] (yellow).

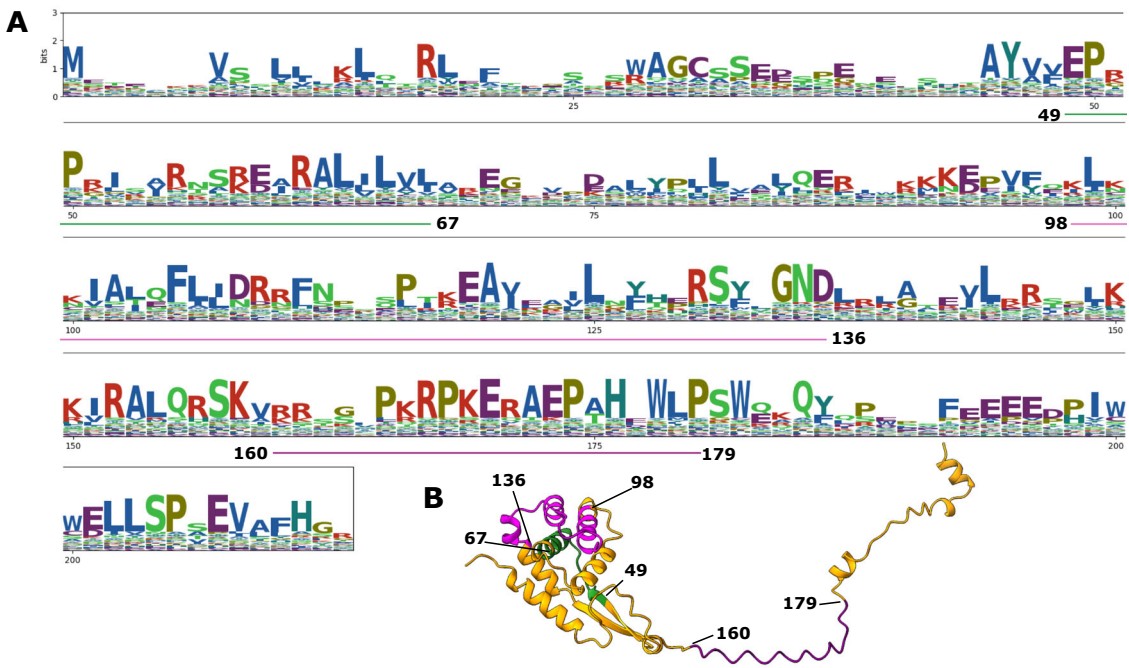

**Fig. 5 | Conserved domains in HsObin-1 protein. A** Amino acid logos (bits) obtained from the alignment of the 44 HsOblin-1-like proteins with conserved stretches underlined in green, pink, and purple. Sites of the alignment matching HsOblin-1-Oi are shown. **B** AF2 HsOblin-1-Oi structure prediction with the respective conserved domains from (**A**) colored accordingly.

bacterial host[4,42]. Here, we expand the diversity and upper temperature limit of Oblin-like cccRNA (79.3 °C).

The discovery of HsOblin-1, combined with the extensive metatranscriptome mining for Oblins reported here, nearly doubled members of this protein family, revealing a degree of diversity that extends beyond sequence similarity, whereas the core protein fold is conserved, albeit with some variation, allowing recognition by structural comparison. Therefore, the 21 major clades identified here (O1-18 and C1-3) can be considered provisional families within the Obelisk superfamily until a proper taxonomy is established. Notably, a long, disordered loop of Oblins showed sequence conservation within individual clades but not across the Oblin protein family, suggesting functional diversification. This diversification also seems to be reflected by the observation that recently published Oblin-1 profiles based on Oblin-1 proteins mined from the SRA[43] can detect only a fraction of Oblins from clades C1, C2, and O18 (below 20%), and if so, only with low confidence (Fig. S14). Oblin-1 proteins from the clade containing the HsOblin-1 were detected at higher frequencies but with a relatively low confidence (more than 50% with e-value above $10^{-5}$ and 100% with e-values above $10^{-8}$, the final cut-off applied in this study to confidently identify Oblin-1 proteins). This observation underscores the crucial role of structural comparisons to confirm or reject potential homology for low-confidence hits.

We did not detect broadly conserved Obelisk-encoded proteins other than Oblin-1. In particular, Oblin-2 predicted in the original Obelisk description was detected only sporadically in closely related Obelisks, casting doubt on the actual expression and functionality of this protein. However, in several new subfamilies of Obelisks, conserved small proteins with confidently predicted simple, primarily, alpha-helical folds were identified. No homologs outside Obelisks were detected for these predicted proteins, which seems to be a common situation for alpha-helical encoded in RNA genomes[44]. Thus, it appears that Obelisks as a class encode a single protein, Oblin-1, which is, in all likelihood, essential for their replication. Other than Oblin-1, some groups seem to have evolved additional protein-coding genes, possibly encoding proteins involved in host-specific interactions.

The Oblin-1 homologs were detected in metatranscriptomes from a broad variety of environments, suggesting that Obelisks replicate in diverse bacteria, although specific hosts could not be readily identified apart from the original observation of Obelisk replication in *Streptococcus sanguinis*. For the HsObs identified in this study, SSU rRNA profiling of the corresponding hot spring metatranscriptomes showed that *Hydrogenobaculum* was the overwhelmingly dominant species, representing more than 95% of bacterial SSU rRNA reads in H5 and the majority (about 67%) in Oi. Notably, archaea constituted a substantial fraction of the microbial community in Oi (15.77%), whereas they were only marginally detected in H5 (0.42%), with only trace levels of other bacteria and less than 1% eukaryotic reads[30]. This ecological pattern indicates that HsObs most likely replicates in *Hydrogenobaculum* cells, although additional evidence will be required to validate this host assignment. In addition, high confidence CRISPR spacer matches identified gastrointestinal bacteria (e.g., families *Lachnospiraceae* and *Selenomonadaceae*) as likely hosts for Obelisks, extending the original observations on Obelisk replication in *Streptococcus sanguinis*, an oral commensal bacterium[23]. These findings, together with the recent demonstration of the high abundance of Obelisks in the oceans[24], show that these are major components of the global microbiome that completely escaped detection prior to the recent advances of metatranscriptome mining.

The lifestyle(s) of the Obelisks and other cccRNA remains a major unresolved issue. Are these RNA plasmids or infectious agents, and to what extent do their replication strategies vary? These questions, as well molecular mechanisms of Obelisk replication, remain to be explored through the identification of the hosts and laboratory cultivation.

## Methods
### Plant material
Chrysanthemum morifolium plants were grown from seeds of the commercial line "Nihon Kogiku" (mixture) obtained from Sakata Seed Corporation (Yokohama, Japan) and maintained in a greenhouse at the National Agriculture and Food Research Organization (NARO). Upper

fully expanded leaves infected with Chrysanthemum stunt viroid (CSVd) were collected.

## FLDS analysis for plant leaves

Chrysanthemum Stunt Viroid (CSVd)-infected chrysanthemum leaves (0.2 g) were crushed using a mortar and pestle in the presence of liquid nitrogen, and total nucleic acids were extracted. After obtaining the dsRNA fraction using cellulose resin, residual non-dsRNA was degraded by treatment with DNase I and S1 nuclease. The purified dsRNA was fragmented using ultrasound (Covaris S220; Woburn, MA, USA), followed by ligation of the U2-primer to the 3′ end of the RNA. Reverse transcription was performed using the SMARTer system (Takara Bio, Kusatsu, Japan) with a primer (U2-comp-primer) containing the complementary sequence. After amplifying the cDNA by PCR, an Illumina sequencing library was prepared using the KAPA HyperPrep kit. Sequencing was carried out using Novaseq (Illumina) through an external company. Details are shown in Supplementary Methods.

## Summary of dsRNA metatranscriptomes from geothermal hot springs in Japan

The hot spring dsRNA metatranscriptomes were constructed using the FLDS from high-temperature acidic hot spring microbiomes in Kyushu Island, Japan, as described previously (Table 1) (see Supplementary Methods)[30]. The temperature and pH ranges for the samples ranged from 68.7 to 95.9 °C and pH 2.2 to 3.7, respectively. Basic geochemical parameters of the hot spring samples, including temperature, pH, electrical conductivity, major ions, and dissolved silica, were obtained from previously published measurements of the same sampling sites. Microbial community structures were determined by SSU rRNA profiling using phyloFlash[45], which showed that *Hydrogenobaculum* (*Hydrogenobaculaceae*) overwhelmingly dominates the bacterial community in H5 and represents the majority taxon in Oi. A summary of these geochemical measurements and SSU rRNA compositions is provided in Supplementary Table S2.

## Data processing

Trimmed reads were obtained using a custom Perl pipeline script (https://github.com/takakiy/FLDS) from dsRNA raw sequence reads[25]. The clean reads were subjected to de novo assembly using SPAdes[46] genome assembler v3.15.5 with metaplasmid mode. The sequences in the output file "before_rr.fasta" were used for the ccfind program (https://github.com/yosuken/ccfind) to find circular sequences with the following parameters: -L 130 -l 30. Sequences with similarity meeting the BLAST threshold ($E$-value $\leq 1 \times 10^{-05}$) were filtered out by BLASTn search against the NCBI nt database, and BLASTx search against the nr database, using an $E$ value threshold of $\leq 1 \times 10^{-05}$, with all other parameters kept at the NCBI BLAST default settings. In addition, sequences with low read coverage (<10 reads), short lengths (<200 nt), or insufficient secondary structure stability (base-pairing proportion ≤60% at 37 °C), calculated using RNAfold[40,47] were removed. To further select contigs with uniform and unbiased read distribution, we calculated two metrics: normalized coefficient of variation (CV) of read coverage and normalized entropy of read start positions. The normalized CV was calculated as the standard deviation divided by the mean of the read depth at each nucleotide position, and further normalized by the theoretical maximum CV expected for a sharp, single-peak coverage. This metric allows for fair comparison across sequences with different lengths and coverage depths. In contrast, normalized entropy was calculated based on the distribution of read start positions along each contig. Shannon entropy was first computed from the frequency distribution of start sites, then normalized by the maximum entropy possible for the given sequence length. This reflects the degree of randomness and diversity in where sequencing reads begin. Only contigs with a normalized CV < 0.6 and normalized entropy >0.7 were retained for downstream analyses.

## Search for CRISPR spacer sequences matching HsObs

In a previous study[30], we obtained small-scale (3.1 Gbp of raw sequence reads in total) metagenome sequences from four samples (H4, H5, Oi, and Y66). Reads from these four samples were assembled de novo, sample-by-sample, using MEGAHIT ver. 1.1.4, after quality filtering using Trimmomatic ver. 0.35, as detailed in a previous study[48]. CRISPR arrays were predicted using minced v0.4.2 (https://github.com/ctSkennerton/minced). This identified 152 CRISPR arrays, including 720 spacers. In addition, a collection of 40,704 *Sulfolobales* spacers from Beppu hotspring[36] was used to find matches in HsObs. Spacer matches (≥30 bp) were not identified for HsObs, using blastn with options "-word_size 7 -dbsize 100000000 -evalue 0.1".

## Structure prediction, comparison, and visualization for HsOblin-1-Oi protein

Initial structural predictions for HsOblin-1-Oi and Obelisk-alpha Oblin-1 were performed using ColabFold 1.5.1 installed locally through Local-ColabFold (https://github.com/YoshitakaMo/localcolabfold) with default MSA setting and alphafold2_multimer_v3 as --model-type[34,49]. The number of recycles used for the resulting models of HsOblin-1-Oi (pLDDT = 84.5, pTM = 0.62) and Obelisk-alpha Oblin-1 (pLDDT = 79.7, pTM = 0.651) were 20 and 5, respectively. The MatchMaker tool of ChimeraX was used to align and compare the predicted structures of HsOblin-1-Oi and Obelisk Oblin-1 proteins. The Oblin-1 model was aligned to the confidently predicted α helical regions (amino acid 51–148) of HsOblin-1-Oi. The RMSD value for all aligned pairs was 3.817 Å. For comparison, the "best" Foldseek hit A0A481Z095 ($E$ value 0.31) showed the RMSD of 12.670 Å when aligned to the HsOblin-1-Oi aa 51–148 model by the MatchMaker tool. AlphaFold3 predictions for protein-alone and RNA-protein complex of HsOblin-1-Oi and Obelisk-alpha Oblin-1 were performed using DeepMind's AlphaFold server (https://alphafoldserver.com/[35]). Model display, structural alignment, coloring, and figure preparation were performed using UCSF ChimeraX software[50].

## Screening public metatranscriptomes for HsOb-Oi/H5-related sequences

Metatranscriptomic datasets were mined from the NCBI Sequence Read Archive (SRA) by querying the taxonomic term "hot spring metagenome (txid433727)" together with the keyword "RNA". The search, performed on 31 March 2025, returned 247 sequencing experiments. Datasets generated on non-Illumina platforms, those with insufficient read depth, and entries previously submitted by our group were discarded, leaving 154 samples corresponding to 197 sequencing runs (Supplementary Data 1). For each run, raw reads were screened against the HsOb-Oi and HsOb-H5 reference sequences using BLASTN (BLAST+ v2.13.0, default parameters); no matches meeting the $E$-value threshold were detected. The open reading frames (ORFs) of HsOb-Oi and HsOb-H5 were then translated and used as a database in BLASTX searches of the same read sets ($E$ value $\leq 1 \times 10^{-5}$). This protein-level search identified eight samples (nine runs) that contained homologous sequences. Reads from these eight samples were assembled de novo in a sample-by-sample manner with MEGAHIT ver. 1.1.4, after quality filtering using Trimmomatic ver. 0.35, as detailed in a previous study[48]. Contigs of each assembly were re-screened with BLASTX using the same $E$ value threshold. Five contigs showing similarity meeting the $E$ value threshold ($E$ value $\leq 1 \times 10^{-5}$) to the HsOb-Oi/H5 ORFs were extracted. Of all the sequences, only one was successfully recovered as a circular sequence using ccfind, and the sequence was circularly permuted not to cut off an ORF in the middle. Notably, all five contigs originated from samples of Yellowstone hot springs.

## Search for Obelisks and Oblin-1 homologs in the putative cccRNA dataset

To identify additional Oblins and Obelisks related to HsOblin-1 and classical Oblin-1, about 4.8 million putative proteins (at least 60 aa long) from 8.9 million recently published[19] putative cccRNAs from more than 5000 metatranscriptomes were searched for the presence of HsOblin-1 and Oblin-1 homologs using an iterative search procedure. One round of PSI-BLAST (version 2.17.0+)[51] was run with the two aligned HsOblin-1 sequences as the query, and the matches meeting the $E$ value threshold ($E$ value $\leq 1 \times 10^{-05}$) were aligned with MUSCLE5 (version 5.1)[52] to build a HMMER profile. The same procedure was followed for a prototype Oblin-1 (Obelisk_000001_000001_000035 from Zheludev et al.) to fetch homologous Oblin-1 proteins and the respective cccRNAs. The resulting HsOblin-1 and Oblin-1 HMMER profiles were enriched by an HMMER hmmsearch run (version 3.4, http://hmmer.org/) against the 4.8 million proteins derived from putative cccRNAs and against 1700 centromer Oblin-1 proteins recently published[19,23]. All confident HMMER hits ($E$ value $\leq 1 \times 10^{-10}$) were kept, aligned with MUSCLE5, and a final HMMER profiles build. For HsOblin-1, all 31 sequences stem from metatranscriptomics. For Oblin-1, 456 sequences stem from metatranscriptomics, and 1181 from centroids, and three individual alignments and profiles were constructed using either only Oblin-1 sequences from metatranscriptomics or centroids or all combined. All final profiles were run again against the 4.8 million proteins from metatranscriptomics-derived cccRNAs (HMMER hmmsearch, default parameters). All about 2600 HMMER hits for HsOblin-1-like and Oblin-1 were collected independent of their $E$-value, to allow discovery and inclusion of distantly related Oblin-1 proteins. This assemblage was enriched with about 1700 Oblin-1 centroid proteins identified by Zheludev et al.[23], the two HsOblin-1 proteins identified by FLDS, and five (one full-length and four partial sequences) HsOblin-1 homologs identified in Yellowstone hot spring sequencing projects (see above). Proteins were subsequently deduplicated (100% identity), leading to 3454 deduplicated (Hs)Oblin-1 homologs.

This protein sequence set was analyzed to identify sequences that could be used as seeds for additional Oblin1-like profiles, as well as to identify false-positives among the hits with high $E$ values. To these ends, proteins were clustered using MMSEQS2[38] (Version: b22d5f6d02cb27ebc2cd931d8d20fe92ff54b8a8, 0.5 sequence identity, 0.7 coverage, resulting in 1211 clusters). The sequences in each cluster were aligned using MUSCLE5[52], and an HMM profile was built for each alignment using HH-suite3[53]. In addition, protein structures were predicted for 1211 representative sequences from the initial MMSEQS2 clusters using AlphaFold3 (see below). Clusters were compared to each other using HHSEARCH (version 3.0.3)[54] and most related clusters aligned to each other by HHALIGN (version 3.0.3)[54] as previously described[55]. In brief, sites in each cluster alignment containing more than 67% gaps were temporarily removed and pairwise scores obtained from an all-vs-all comparison with HHSEARCH. Pairwise scores $d_{A,B}$ of cluster A and B were converted into distances using the formula $d_{A,B} = -\log[S_{A,B}/\min(S_{A,A}, S_{B,B})]$ in which $S_{A,B}$ is the HHSEARCH score between the respective pair and $S_{A,A}$ and $S_{B,B}$ the self-score. Matrix of pairwise scores was used to construct a UPGMA dendrogram using the R package hclust (1.1) with the argument "method" set to "average" (=UPGMA). Clusters corresponding to related, shallow tips of the dendrogram were iteratively aligned to each other using HHALIGN. Further, neighboring clusters were only aligned if (i) the pairwise distance did not exceed 2.2 or (ii) the alignment coverage was at least 0.66. The procedure was iterated for 7 cycles, after which merging of alignments decreased the alignment quality (visual inspection). This procedure resulted in 153 clusters of aligned sequences, of which 140 contained less than 10 (spanning 261/3454 sequences). Aligned clusters were converted into HMMER profiles, which were used as queries to search against the complete set

of cccRNA-encoded proteins to identify clusters that could be used to identify additional, distantly related Oblin-1 homologs. Sequences in clusters that initially contained fewer than 10 sequences were re-aligned with confident hits ($E$ value $\leq 1 \times 10^{-08}$), HMMER HMMs were built and re-run against the full cccRNA-encoded protein set. All additional hits not present yet in the initial assemblage with an $E$ value below $1 \times 10^{-08}$ were kept.

Structure comparison as well as cluster-specific pairwise HMM comparison indicated that 75/153 aligned clusters with 10 or fewer sequences were likely false positives and were accordingly discarded (86 sequences across the 75 clusters, mainly, singletons). Additional distant Oblin1-like sequences not covered by the first original HsOblin-1 and Onlin-1 profiles were detected for 32 of the aligned clusters. Overall, the enrichment runs resulted in an additional 1750 deduplicated Oblin1-like proteins and five clusters, including one spanning HsOblin-1 proteins, not containing known Oblin-1 centroid proteins. These five clusters were aligned, and additional HMMER profiles were built. To find additional Obelisks, over 2,000 assembled metatranscriptomics published after the study of Lee et al.[19] were searched for cccRNAs (see below, about 4.1 million unique cccNAs), ORFs extracted, and 2.2 million proteins of at least 60 aa searched with the different Oblin-1 HMMER profiles. This resulted in 1467 additional unique (Hs)Oblin1-like proteins.

To analyze the overall relationships across all identified 5009 unique (Hs)Oblin-1 homologs, the FLDS identified HsOblin-1-Oi and -H5, the five Oblin-1 proteins identified in a Yellowstone hot spring metatranscriptome, and the 1700 known centroid Oblin-1 proteins. MMSEQS2 clustering and the above-described iterative pairwise alignment of clusters using HHSEARCH and HHALIGN were repeated. Finally, 111 aligned clusters were obtained. The phylogeny for each cluster was constructed using FastTree[56,57] (version 2.1.4, -wag -gamma options)- after removing sites with more than 90% gaps from the respective MSAs. The obtained trees were midpoint rooted and grafted to the HHSEARCH-based UPGMA dendrogram of the aligned clusters. Protein structures were predicted for a representative set of diverse sequences across each clade using AlphaFold3 (see below for details).

### Tree visualization

Phylogenetic trees and dendrograms were visualized with iTol web interface (https://itol.embl.de/)[58].

### Search for additional cccRNAs in metatranscriptomes

Assembled metatranscriptomics deposited after the publication of Lee et al.[19] were retrieved from IMG/MER[37] (until May 6th, 2025), and putative cccRNAs were retrieved with vdsearch[19] (https://github.com/Benjamin-Lee/vdsearch version 0.0.1). Monomeric cccRNAs were further dereplicated with circKit (https://github.com/Benjamin-Lee/circkit version 0.1.0).

### Prediction of protein-coding genes

ORFs on cccRNAs were identified and predicted in all 6 frames by circKit (https://github.com/Benjamin-Lee/circkit) default settings (start-stop, accounting for circularity and stopping after three wrappings of the genome in case no stop is present). For the Oblin-1 search, only putative proteins of at least 60 aa were considered. For non-oblin-1 proteins, all proteins of at least 40 aa from Obelisks encoding an (Hs) Oblin1-like protein were considered.

### Non-Oblin-1 protein analysis

Putative proteins of at least 40 aa encoded on Obelisks besides Oblin-1 (named here non-Oblin-1 proteins) were retrieved from all six frames and clustered with MMSEQS2[38] (default parameters except: min-seq-id 0.4 -c 0.7). For clusters with at least two members, protein structures were predicted with AlphaFold3 (version 3.0.1, see below). Structures

were clustered with foldseek[59] (version: d2d09b588f50d5f8e2f-d7a958377a33b2f725415, coverage -c 0.8). Structure-based clusters were inspected manually, indicating that clustering is mainly based on small, simple secondary folds such as single alpha-helices. Therefore, to avoid clustering of unrelated proteins, only sequence-based clustering was considered further. Sequence-based clusters with more than 5 members were aligned with MUSCLE5[52] and compared with HHBLITS (two iterations) against the following broad HHSUITE databases: pdb70[60], pfam[61], scope[62,63], ECOD[64]; and two HHSUITE databases for RNA virus proteins: nvpc[65] and viral[66]. Only 4/18,609 clusters showed hits with a probability higher than 90, and these 4 hits were mainly mapping back to short helices in the non-Oblin-1 proteins, likely unrelated to the function associated with the hits. Further, all putative non-Oblin-1 proteins (independent of the sequence-based clustering) were searched against the HMMER pfam database (based on pfam34.0), retrieving no conclusive hits.

Sequence-based clusters containing at least 20 proteins were projected on the Oblin-1-based tree. The deepest node being parental to all Obelisks encoding one non-Oblin-1 protein of the respective cluster was identified, and the fraction of leaves encoding for a cluster-specific non-Oblin-1 protein in this clade was retrieved. Further, the mean pairwise distance of all leaves in this clade was retrieved. We excluded clades with a mean pairwise distance of less than 0.1 to avoid the detection of false-positive non-Oblin-1 ORFs by the high similarity of the underlying Obelisk sequences. Further, clades with a coverage of at least 70% of leaves encoding for the putative non-Oblin-1 proteins were inspected in detail. In addition, the two clades with more than 100 putative non-Oblin-1 proteins were also inspected, although providing a lower coverage. This retrieved 12 sets of putative non-Oblin-1 proteins, which are evolutionarily related to the respective Oblin-1 protein. Protein structures and ORF positions were visually inspected for these 12 clades. Further, the percentage of predicted base-pairing between the non-Oblin-1 and the Oblin-1 ORF in the secondary structure prediction (both sense and antisense) was extracted from the RNAfold-based prediction (see below).

In addition, the full non-Oblin-1 protein set was searched for Oblin-2 proteins described previously by Zheludev et al. Therefore, a prototype Oblin-2 sequence ("MDS VQI LRK KIL KNE EQR EFL LKK IGN LEY EIN NLE HKI ENQ QRV LQN LLR EK")[23] was used as PSI-BLAST query and run until convergence (default settings except: -evalue 0.01 -outfmt 6 -max_target_seqs 50000 -num_iterations 0). Putative Oblin-2 ($E$ value $\leq 1 \times 10^{-03}$) were mapped on the Oblin-1 phylogeny.

Non-Oblin-1 proteins were inspected for the presence of leucine zipper motifs. Coiled-coil regions were predicted with a local version of CoCoNat[41] (default settings). Coiled-coil regions of at least 21 amino acids were further inspected for the presence of leucine repeats ($i + 7$ heptad repeats (abcdefg) in which "a" and "d" positions are hydrophobic amino acids ("a" leucine, "d" leucine or isoleucine[67])). Leucine zippers are reported if at least 3 heptad repeats (21 amino acids) contain in 50% of the cases a leucine at the "a" position of the heptad and either leucine or isoleucine at the "d" position and at least three matches either at the "a" or the "d" position.

### Structure prediction, comparison, and visualization for the expanded Oblin1-like protein family

Protein structures for the expanded (Hs)Oblin1-like proteins and non-oblin-proteins were predicted with a local version of AlphaFold3[35] (version 3.0.1). Besides default settings (--seeds 1,2,3,4,5), all dereplicated putative proteins from 8.9 million cccRNAs found in metatranscripomes[19] (majority are non-Obelisks) have been visible to AlphaFold3 during the MSA construction phase by concatenating them to the uniref90 file used by AlphaFold3 (uniref90_2022_05.fa). Chimera X (version 1.3)[50] was used to visualize structures.

Selected (Hs)Oblin-1-like core structures (globular fold surrounded by the conserved beta sheets) have been compared pairwise (Fig. S2) using the Dali web server[68]. The pairwise Dali translation-rotation matrix coordinates were used in Chimera X to superimpose structures using the "view matrix mod" command.

Foldseek easy-search[59] was used to search non-Oblin-1 proteins against several foldseek databases (pdb, alphafold2 proteome, alphafold2 swissport, and big fantastic database). Foldseek search (-c 0.8) and clust (default) commands were used to compare non-Oblin-1 proteins against each other.

### RNA secondary structure prediction for Obelisks from metatranscriptomic data

RNA secondary structures of metatranscriptomic-derived Obelisks and centroids were predicted with RNAfold implemented in the ViennaRNA package (version 2.5.1)[40], accounting for circularity (RNAfold -c) and predicting structures for both the sense and the antisense strand. For each structure, the minimum free energy (mfe) as well as the percentage of paired bases was extracted. Further, the fraction of paired bases for which both bases reside within the Oblin-1 ORF among all paired bases for which at least one member resides inside the Oblin-1 ORF was extracted from the dot bracket secondary structure annotation of the RNAfold output.

### cccRNA genome visualization and Jupiter plots

Obelisk maps were plotted using the Python library Plasmidviewer (https://github.com/ponnhide/plasmidviewer, version 0.0.0), which takes the coordinates from a GenBank formatted file. Obelisks were oriented in a way that the Oblin-1 ORF is always visualized on the sense strand. Jupiter plots of RNA secondary structure base pairs were plotted using the Python script "jupiter" (https://github.com/rcedgar/jupiter, downloaded June 2025), which takes the base pairing coordinates from the RNAfold RNA secondary structure prediction.

### Ribozyme prediction

To detect ribozymes among the Obelisks detected in metatranscriptomic data, infernal (http://eddylab.org/infernal/ and ref. 33) as implemented in vdsearch[19] (version 0.0.1, infernal version 1.1). Running vdsearch infernal, default settings with a static ribozyme collection from ViroidDB[69] (see supplementary file "infernal/ rfam_cm" for covariance models).

### CRISPR spacer search

2x concatenated Obelisk sequences (both sense and antisense) have been used to search a local CRISPR spacer database, for which all CRISPR Arrays in all bacterial and archaeal genomic sequences in RefSeq as of May 2023 were predicted using minced (https://github.com/ctSkennerton/minced). BLASTN was used (blastn -query ip_fasta -task blastn -db sp2305.db -out op_file -evalue 1000 -word_size 4 -outfmt "6 qaccver saccver pident length mismatch gapopen qstart qend sstart send evalue bitscore nident slen"). Spacers covered by at least 90% and at least 16 matching nucleotides have been inspected further.

Further, high quality spacers of the JGI spacer db[70] (https://spacers.jgi.doe.gov/quick_start/; nr_spacers_hq-all_25-05-10.fna.gz) have been searched with 2x concatenated Obelisk sequences (both discovered here and published) using MMSEQS2. First, an MMSEQS2 database was created for both the spacers and the Obelisks (mmseqs createdb ip_fila op_db). Then, the Obelisk db was searched against the spacer db with specific parameters (mmseqs search obelisk_db spacer_db obelisk_spacer tmp --min-seq-id 0.8 --cov-mode 1 -c 1.0 --alignment-mode 3 -s 8.5 --max-seqs 1000000 -e 0.001 --search-type 3) and the hits converted into a tabular output (mmseqs convertalis obelisk_db spacer_db obelisk_spacer op_table.m8). Spacer cluster hit characteristics were retrieved from the adjacent JGI CRISPR SQL duckdb

(global_crispr_db_full_2025-05-02.duckdb), including associated CRISPR types and putative hosts. Spacer hits were then mapped to the respective Obelisks, and spacers combined if their non-overlapping sequences were 10 or fewer nt long. Putative hosts associated with the spacers were retrieved, and 404 reference genomes of the same families from NCBI GenBank in November 2023 (prok2311) were inspected for the presence of a reverse transcriptase annotation within the respective CRISPR loci.

## Visualization of sequence conservation

Visualization of sequence conservation of aligned Oblin-1 clusters as stacked sequence logos (information content as bits[71]) was done with the Python package logomaker[71].

## Obelisk clustering

To follow the nomenclature proposed by Zheludev et al., we clustered all known Obelisks (~7000 from Zheludev et al.[23] and 40 from López-Simón et al.[24]) with the ones discovered here at 80% nucleotide sequence identity (ANI) with circuclust (https://github.com/rcedgar/circuclust version v1.0.i86linux64). Cluster members were then individually clustered at 95% ANI to follow the nomenclature: "Obelisk_X_Y_Z," "X" denotes no. of clusters at 80% nucleotide identity, "Y" at 95%, and "Z" as the identifier for all strains within a given species.

## "Logan" profile comparison

In order to assess whether recently published Oblin-1 HMMs[43] could, in principle, detect our newly identified Oblins, HMM profiles for Oblin-1 from the Logan depository (all Oblin-1 HMMs deposited in s3://logan-pub/paper/Obelisk/12_Obelisk_Logan_Run2) were retrieved and run with HMMER (hmmsearch, default parameters) against the Oblin-1 proteins identified in this study. E-values were collected for each Oblin-1 protein and mapped back to the respective Oblin-1 clades.

## Reporting summary

Further information on research design is available in the Nature Portfolio Reporting Summary linked to this article.

# Data availability

All data generated or analyzed in this study are publicly available. Sequence data have been deposited in the NCBI Sequence Read Archive under accession numbers DRR751250 (CSVd), DRR460909–DRR460930 (HsObs), and DRR898023–DRR898026 (hot spring metagenomes). Previously reported sequencing datasets screened in this study include publicly available hot spring metatranscriptomes retrieved from the NCBI Sequence Read Archive (SRA) (listed in Supplementary Data 1), the metatranscriptomic cccRNA dataset reported by Lee et al.[19], and additional assembled metatranscriptomes retrieved from the IMG/MER system[37]. CRISPR spacer datasets were obtained from a local RefSeq-based spacer database (https://github.com/ctSkennerton/minced) and from the JGI spacer database (https://spacers.jgi.doe.gov/quick_start/; nr_spacers_hq-all_25-05-10.fna.gz). Additional data, including identified Obelisk sequences and their encoded proteins, are available without restriction via Zenodo (https://doi.org/10.5281/zenodo.16965620). Tables for individual graphs are summarized in a Source data directory (Source_data_fig.zip), which is included in the Zenodo repository. Source data are provided with this paper.

# Code availability

The custom scripts used in this study are publicly available via Zenodo with assigned DOIs: https://doi.org/10.5281/zenodo.18616998 (FLDS pipeline)[72] and https://doi.org/10.5281/zenodo.18575462[73]. The corresponding development repositories are maintained on GitHub (https://github.com/takakiy/FLDS and https://github.com/yosuken/ccfind).

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

## Acknowledgements

The authors gratefully acknowledge NARO GenBank for providing viroid strains used in this study. The authors thank Hiroko Mizukoshi and Tamaki Ichiki-Uehara for their experimental and technical support. This

study was supported by Grants-in-Aid for Scientific Research on Innovative Areas from the Ministry of Education, Culture, Science, Sports, and Technology (MEXT) of Japan (Grant Nos. 25K22486 [for S.U., Y.M., and Y.N.], 23H18146 [for S.U. and Y.M.], 24K02083 [for S.U.], and 20K20377 [for T.N.]). This work was also supported by the JST FOREST Program, Grant Number JPMJFR240T [for S.U.]. This research was also supported in part by Lilly Endowment, Inc., through its support for the Indiana University Pervasive Technology Institute, which provided supercomputing resources for protein structure modeling. This work utilized the computational resources of the NIH HPC Biowulf cluster (https://hpc.nih.gov). The authors appreciate Igor Tolstoy's (NCBI, NIH) help in the construction of the local CRISPR spacer database used in this work. The authors thank Benjamin Lee for his help with vdsearch and circkit. P.M. and E.V.K. are supported by the Intramural Research Program of the National Institutes of Health (NIH). The contributions of the NIH authors are considered Works of the United States Government. The findings and conclusions presented in this paper are those of the authors and do not necessarily reflect the views of the NIH or the U.S. Department of Health and Human Services.

## Author contributions

All authors made substantial contributions to this work. S.U., A.F., P.M., M.K., E.V.K., and T.N. were responsible for the design of the work, the acquisition, analysis, and interpretation of data, and drafted the initial work. S.U. and Y.M. performed experiments, analyzed, and interpreted the data. S.U., A.F., P.M., M.K., E.V.K., and T.N. substantively revised the work. S.U., A.F., P.M., Y.N., Y.T., S.M., and M.K. performed bioinformatic analysis. S.U., A.F., P.M., Y.M., Y.T., Y.N., S.M., M.K., E.V.K., and T.N. wrote the manuscript.

## Competing interests

JAMSTEC holds a patent related to a method for fragmentation of double-stranded RNA used in the FLDS protocol (WO2017065194A1; EP3363898B1; CN108513581B; JP6386678B2; US10894981), with S.U. and T.N. listed as inventors. The patent covers technical aspects of the dsRNA fragmentation process and does not impose restrictions on the use of FLDS-derived data or on the publication and reuse of the data reported in this study. All data generated in this work are freely available as described in the "Data Availability" statement. The authors declare no other competing interests.

## Additional information

[1]Laboratory of Biology for Extreme Molecule, Department of Life and Environmental Sciences, University of Tsukuba, Tsukuba, Japan. [2]Tsukuba Institute for Advanced Research (TIAR), Microbiology Research Center for Sustainability (MiCS), University of Tsukuba, Tsukuba, Japan. [3]Department of Biology and Department of Molecular and Cellular Biochemistry, Howard Hughes Medical Institute, Indiana University, Bloomington, IN, USA. [4]Computational Biology Branch, Division of Intramural Research, National Library of Medicine, Bethesda, MD, USA. [5]Institute for Plant Protection, National Agriculture and Food Research Organization (NARO), Tsukuba, Japan. [6]Super-cutting-edge Grand and Advanced Research (SUGAR) Program, Japan Agency for Marine Science and Technology (JAMSTEC), Yokosuka, Japan. [7]Cell Biology and Virology of Archaea Unit, Institut Pasteur, CNRS UMR6047, Université Paris Cité, Paris, France. [8]Research Center for Bioscience and Nanoscience (CeBN), JAMSTEC, Yokosuka, Japan. [9]These authors contributed equally: Akihito Fukudome, Pascal Mutz. ✉e-mail: urayama.shunichi.gn@u.tsukuba.ac.jp

