## [Transparent Peer Review file · Nature Communications]

Identification of hot spring Obelisk-like RNA replicons and expanded diversity of the Obelisk superfamily

Corresponding Author: Dr Syun-ichi Urayama

Version 0:

Reviewer comments:

Reviewer #1

(Remarks to the Author)
Review summary

Fukudome, Mutz and colleagues have written a timely and important contribution to the nascent field of Obelisk biology. Their finding of a distinct type of hot-spring associated Obelisk, whose Oblin-1 homologue escaped previous detection using established methods, is particularly striking and exemplifies the breadth of entities yet to be discovered and describes methods competent for such detection. The authors' expansion of known Obelisks and subsequent construction and analysis of dendrograms and phylogenies provides needed contextualization for this genetic system and sets the stage for future work - particularly into environment-x-Obelisk and non-Oblin-1-x-Obelisk association studies.

The manuscript itself is well written and largely easy to follow without major logical leaps in the authors' conclusions. Two major areas stand out as needing attention:

1. Throughout the manuscript (and in these authors' previous papers) the authors refer to these and related entities as "covalently closed circular RNAs" (cccRNAs). However, metatranscriptomic sequencing, including the herein described FLDS method, cannot unambiguously distinguish between bona fide covalently closed and circular RNAs and concatameric sequence repeats. Put differently, the terminal repeat assembly feature used in this (and other) study(ies) to infer circularity (with ccfnd) is as equally consistent with an RNA being circular as it is with that RNA being repetitive. As such, the use of "cccRNA" as a term is somewhat misleading as it draws a conclusion we cannot yet experimentally support. I would request that the authors add a section early on in the manuscript explaining this caveat and that they alter the language in the text to refer the apparent/predicted circular nature of the entities in question.

2. Contrary to the well-written bioinformatic methods sections, the experimental methods sections are written with substantial missing information. A replication study of this manuscript's experimental components would simply not be possible with the information provided. Please rewrite the experimental methods section thoroughly such that a separate lab could repeat this study.

In addition to these main points, the authors use the word "significant" throughout the manuscript and only occasionally does this actually refer to a statistical hypothesis test - please alter your language to reflect conventional scientific use of this adjective.

Overall, I believe the authors should be proud of their work. With these two main points addressed, I believe that this work would be ready for publication. As such I am happy to accept this work with minor revisions, and I thank the authors for their contribution.

Attached are my line-by-line comments, most of which are stylistic or editorial and can be considered as the authors/editors see fit.

Abstract

Overall, the abstract effectively summarizes the work. A few clarifying points are requested.

Line 34: the covalent-closed-nature of a putative circular RNA cannot be inferred from RNA sequencing alone (including FLDS), neither can the circularity of a putatively circular RNA. The features used for inferring circularity are confounded by concatemeric/repetitive sequences, so the two types of species cannot be distinguished by sequencing. For these reasons, I believe that the “covalently closed circular” phrase needs to be couched in less declarative tones e.g. “apparently covalently closed circular” throughout the manuscript when referring to RNA elements from previous studies, including for the Obelisk paper (which are described as cccRNAs in Line 35 too) - where neither the apparent covalent or circular molecular features of these elements were tested appropriately.

Line 41: “Ch” typo

Line 44: If HsObs HsOblins don’t share “significant sequence similarity” with Oblin-1 then how were these elements recognized as Obelisks (implied in Line 42). I think the authors should a) explain how these elements were identified, and b) give a metric to what they mean by “no significant sequence similarity”

Line 44: in my opinion, “significant” is a sacrosanct word in science that implies statistical hypothesis testing, if such as test was conducted then please state which test was used and what p-value cutoff was used, otherwise, please use a different word to describe the nature of the sequence similarity.

Line 44: which sequence, nt or aa?

Line 45: “Metatranscriptomes” are being queried rather than “Metagenomes”

Line 46: “Substantially” what approximate fold change

Line 48: If the authors need more words in the abstract, I think the line pertaining to the Yellowstone MTX could be instead a clause added to the “substantial expansion” sentence

Introduction

Overall, the introduction is written well and gives a good overview of the field to the reader. Several citations are missing throughout (which have been cited previously but should be repeated as specific points are made). My major concerns with the introduction are 1) the use of “cccRNA” to describe entities of un-validated circularity of covalent structure (such as Obelisks), and 2) the breadth of the introduction. For this second point, which I guess is mostly editorial, I defer up to the authors for how they wish to structure their manuscript, but I think the introduction could be shortened for brevity by making it less of a “review” of the field.

First paragraph: this is a nice introduction to viroids, though perhaps is too much background on viroids vs the subject matter of the manuscript. E.g. the details on viroid genome processing could be omitted for brevity

Second paragraph: this is also good, and could also likely be shortened for brevity due to relevance to the manuscript.

Please also re-add the citations present in the first paragraph for the specific examples described in this second paragraph (see below).

Line 66: “bona fide” should be in italics

Line 67: add “virusoid” citation

Line 69: add “retroviroid” citation

Line 72: add “retrozyme” citation

Line 82: add recent HDV-like citations

Third paragraph: again, a nice intro into HDV but could also be condensed down

Line 90/91: add citation for CRISPR x cccRNA and an approximate count of what “several” means

Line 94: add citations for HDV-like elements of which there are several

Line 95: add citations for ambiviruses

Fourth paragraph: again, this is a nice introduction/summary of ambiviruses but I’m not sure all this detail/context is needed for this manuscript

Main Text

Overall, the main text is written well and is easily understood. Editorially, it is clear that the two first authors each wrote different parts of this section - that is completely fine and expected - but it does mean that the first half doesn’t reference the later “methods” section and the latter half does; which makes my feedback over this section inconsistent. Having next read the “methods” section, some of the points I make here are addressed and could be referenced by stating “see methods” as the authors see fit. Scientifically, everything is well written and clear. I am particularly interested in the finding that the HsObelisks appear to not be detectable by the original Oblin-1 pHMM (though this is not explicitly stated in this section - please make this clearer if this is indeed the case) - which is very cool! Additionally, recent efforts (Logan bioRxiv, Chikhi et al. 2025) have built a new Oblin-1 pHMM and Obelisk database, it would be interesting to see how these compare to your findings but I also do not think this is a required comparison to make, please at least add a citation to this work reflecting the evolving nature of this space. You should probably also cite Edgar et al. 2022 when referencing HDV-like MTX mining.

General point: where possible/known, please list the version numbers of the softwares used

General point: where possible/known, please list the dates of accession to databases (e.g. BLAST) when they were used

Line 120: same comment as before, we cannot determine “circularity” from these data

Line 121: RCR is not the only conceivable mechanism by which a cccRNA might replicate. Could be discontinuous RCR (Jain et al. 2020) or rolling hairpin replication (Tattersal et al. 1976), or other mechanisms. Instead of diving into the weeds of putative replicative mechanisms, I would just explain that RNA-dependent RNA replication requires the generation of a template-substrate dsRNA duplex (regardless of replicative mechanism), which can be captured by dsRNA sequencing

Line 128: add citations to previous methods that have used terminal repeats for the same purpose

Line 129: BLAST doesn’t do significance tests so I would avoid using the word “significant” and instead use “substantial” (or

equivalent) and quote an E-value/BitScore/etc threshold used for this conclusion

Line 131: what units is the "<10" value referring to? Likewise, what units are in the "10x" value?

Line 132: which RNA folding program was used for this prediction and what settings (e.g. if viennaRNA was used, was "circularity" enforced?).

Line 133: this sounds like an interesting analysis, please explain it a little more for the reader and also cite the conclusions from your measurement of the "evenness of mapping" (including units) and any relevant citations too

Line 138: this method cannot differentiate between circles and concatemers.

Line 140: "circular"

Line 145: "significant"

Line 147: coverage units

Line 167: which covariance models / ribozymes were searched - please add Rfam accessions or and/add a supplementary file

Line 171: "significant"

Line 172/173: please add AF2 and AF3 settings and citations. For AF2 please also add how the MSA was constructed (if it was) for structural inference

Line 177: how was the sequence alignment (aa or nt) constructed, and what sequence was this ORF aligned to? Was an alignment against the original Oblin-1 pHMM attempted?

Line 183: "significant"

Line 196: please add how different the pLDDT scores were +/- RNA

Line 202: likewise, please add pLDDT/pAE numbers to this text

Line 215: please state which tool / settings / database was used for the spacer search

Line 254: how is coverage calculated here?

Line 280: how does the RNA secondary structure map to the Oblin-1 ORF, is it similar to other Obelisks where Oblin-1 is predicted to be largely self-complementary?

Line 291: the value of 51 nt from the 5' end is eye-catching because that is the arbitrary indexing convention of the Oblin-1 A(51)UG of the sense strand from the 5' end of the Obelisk genome (50nt upstream sequence, then start of Oblin-1 at 51nt) - please add a note to specify if this median distance is artifactual from this arbitrary convention and/or please clarify if these 51nt are from the STOP of the Oblin-1 in the antisense strand (the language of 5' +/- sense/antisense gets confusing)

Line 293: what frequencies?

Line 320: "significant" and do the other alpha-helical non-Ob-1 ORFs contain leucine zipper motifs?

Discussion

The discussion is well written and aptly summarizes the work without making unsubstantiated claims. I suggest a few edits to qualify some statements.

Line 354: I wouldn't say finding the spacer has "clinched" that that cccRNA has a bacterial host because we still don't really know how well cccRNAs are surveilled by the CRISPR system (or really much about RNA-surveilling spacer acquisition), or have a notion for how reliable spacer-based-host-ID really works, so I would change the language here to reflect that this observation supports the putative host being bacterial.

Line 364: I think it's a little premature to conclude from the hallucinations of AF3 that there's a specific HsOb RNA binding interaction. I would further qualify this sentence to reflect how very preliminary this association is.

Line 377: I think the authors should explain this Hydrogenobaculaceae association more clearly (that the MTX suggests this is the only -?- genus of bacteria in the sample) either here or in the main text

Methods

The bioinformatic methods sections are written well with appropriate details and a lot of the points raised in the earlier main text section are answered here. The experimental methods sections however lack critical detail on almost every point. The authors need to completely re-write the experimental methods sections such that every single step of the protocols carried out can be replicated by another lab. Note, it is also insufficient to cite previous work for methods details.

Line 389: what cultivar? Which leaves? Where was the sample taken from in the leaf and how?

Line 390: what volume of LN2? what method was used for nucleic acid extraction? Buffers, volumes, incubation temperatures, citations, etc..

Line 391: what cellulose-based method was used for dsRNA enrichment? Buffers, volumes, incubation temperatures, citations, etc..

Line 392: where were these enzymes sourced from? What volumes, temperatures, buffers etc. Were the enzymes used simultaneously or sequentially? How were the enzymes removed from the reaction mix prior to library prep? What are the expected nucleic acid depletions from using these enzymes?

Line 392: what settings were used for fragmentation? Was the sample cooled? What volume was used? Etc etc etc

Line 393: what is the U2 primer? Is it cited in the supplemental tables? How was it ligated? How was the material purified etc etc.

Line 394: which kit was used? which enzyme was used, what experimental settings, how long, how was it purified?

Line 395: what PCR cycling conditions were used, what enzyme, what primers, how was indexing done, which indices were used?

Line 396: please provide every detail of the library prep

Line 397: which Novaseq was used, how many reads, was PhiX used as spike-in, what was the read length, what was the sequencing strategy (SE or PE), what was the name of the company?

Line 403: please summarize and then provide these “geochemical and community structure” data - and explain how they were determined.

Line 404: this line appears to be irrelevant to this particular manuscript?

Line 411: please provide a link to ccfind

Line 413: default BLASTn/x settings?

Line 459: I think there’s a typo here, what does “4.8 putative proteins” refer to?

Line 600: There’s a typo throughout the manuscript referring to “Jupiter” plots as “Jupyter” plots - the name refers to the banded structure of the gas giant, not the notebook software.

Line 636: thank you for making your data publicly available.

Figures

The figures are clear and their captions make sense. I would request that the figures are rendered in a higher DPI and time permitting, with larger components and better color contrast for folks hard of sight.

Reviewer #2

(Remarks to the Author)

In this work, the authors explored dsRNA metatranscriptomes constructed by Fragmented and primer- Ligated DsRNA Sequencing (FLDS) , a method for selectively sequencing of dsRNA, a typical replicative intermediates of RNA replicons. Your main text is so difficult to reads a need a significant improvement. It is almost impossible to follow your story. Did you tried to amplify your dataset with other samples ? s not visible from how many samples you started your research

Reviewer #3

(Remarks to the Author)

Uryama and colleagues here present a significant addition to the small but growing number of studies that have identified cccRNAs known as Obelisks. Their work relied on a unique strategy of looking within samples from acidic hot springs and used metatranscriptomes generated by Fragmented and primer-Ligated dsRNA Sequencing (FLDS).

Their results include a novel Obelisk found in the hot springs metatranscriptomes and were further expanded upon by searching for related cccRNAs in a large collection of such sequences. They also identified and characterized several families of novel Obelisk-associated proteins.

These findings should be useful for guiding further identification and characterization of Obelisks and their sequence features, including encoded proteins.

Their manuscript may benefit from considering the following:

- It is not obvious why they specifically chose to look within hot spring environments.
- The significance of predicted structural differences at 37c and 80c could be explored or discussed further.
- The significance of potential RNA-binding regions of HsOblin-1 could be explored or discussed further.
- The lack of CRISPR spacer matching is somewhat surprising given the number of Obelisk sequences that were included in their search, however previous studies have also struggled to find matches. Perhaps more info could be provided on the spacer database searched, and perhaps trying other available databases (eg that from JGI).
- The lack of homologous matches for the non-Oblin-1 proteins despite searching several databases via both profile and structure based comparisons is a bit perplexing, but may just represent the unique novelty of these putative proteins.
- In the discussion they comment on the likelihood of potential RNA binding sites and the Oblin-1 protein being involved in Obelisk replication, but their reported work did not directly address replication.

Version 1:

Reviewer comments:

Reviewer #1

(Remarks to the Author)

I would like to thank the authors for their thorough responses to my reviews. I believe that all of my major points have been addressed - namely the "cccRNA" nuance, and the wet-lab methodological details. I'm especially excited by the more thorough CRISPR spacer analysis - very interesting!

I have no further comments and believe that this exciting work is ready to be published.

Thank you all,

Reviewer #3

(Remarks to the Author)

The authors have addressed all issues identified in the previous review and my recommendation is to accept the manuscript for publication.

Black: Reviewer comments

Blue: Our responses

Reviewer #1 (Remarks to the Author):

Review summary

Fukudome, Mutz and colleagues have written a timely and important contribution to the nascent field of Obelisk biology. Their finding of a distinct type of hot-spring associated Obelisk, whose Oblin-1 homologue escaped previous detection using established methods, is particularly striking and exemplifies the breadth of entities yet to be discovered and describes methods competent for such detection. The authors' expansion of known Obelisks and subsequent construction and analysis of dendrograms and phylogenies provides needed contextualization for this genetic system and sets the stage for future work - particularly into environment-x-Obelisk and non-Oblin-1-x-Obelisk association studies.

The manuscript itself is well written and largely easy to follow without major logical leaps in the authors' conclusions. Two major areas stand out as needing attention:

1. Throughout the manuscript (and in these authors' previous papers) the authors refer to these and related entities as "covalently closed circular RNAs" (cccRNAs). However, metatranscriptomic sequencing, including the herein described FLDS method, cannot unambiguously distinguish between bona fide covalently closed and circular RNAs and concatameric sequence repeats. Put differently, the terminal repeat assembly feature used in this (and other) study(ies) to infer circularity (with ccfnd) is as equally consistent with an RNA being circular as it is with that RNA being repetitive. As such, the use of "cccRNA" as a term is somewhat misleading as it draws a conclusion we cannot yet experimentally support. I would request that the authors add a section early on in the manuscript explaining this caveat and that they alter the language in the text to refer the apparent/predicted circular nature of the entities in question.

We thank the reviewer for this important clarification. We fully agree that FLDS-based metatranscriptomic assembly cannot discriminate bona fide covalently closed circular RNAs from concatameric repeat assemblies, and that strict experimental evidence for covalent circularity is not available. In response, we have added a dedicated paragraph early in the Introduction that explicitly explains this caveat and clarifies that "cccRNA" is

used as an operational term for convenience and readability only. We have also adjusted wording throughout the manuscript accordingly. Please see L107-114 (in the revised manuscript).

2. Contrary to the well-written bioinformatic methods sections, the experimental methods sections are written with substantial missing information. A replication study of this manuscript's experimental components would simply not be possible with the information provided. Please rewrite the experimental methods section thoroughly such that a separate lab could repeat this study.

We thank the reviewer for this comment. To address this concern, we have added a comprehensive Supplementary Methods section that provides a detailed description of the FLDS experimental workflow. This section includes all essential procedures and conditions required for independent replication of the experiments. We believe that this revision resolves the issue and substantially improves the reproducibility of the study.

In addition to these main points, the authors use the word “significant” throughout the manuscript and only occasionally does this actually refer to a statistical hypothesis test - please alter your language to reflect conventional scientific use of this adjective.

We thank the reviewer for pointing this out. In the revised manuscript, we replaced the adjective “significant” with more precise expressions such as “meeting the E-value threshold” or “detectable similarity,” to avoid any unintended implication of statistical hypothesis testing. ‘Significant’ was kept only in those instances where statistical tests are explicitly mentioned. These modifications involved no changes were made to the scientific content.

Overall, I believe the authors should be proud of their work. With these two main points addressed, I believe that this work would be ready for publication. As such I am happy to accept this work with minor revisions, and I thank the authors for their contribution. Regards, Ivan Nikolay Zheludev.

Attached are my line-by-line comments, most of which are stylistic or editorial and can be considered as the authors/editors see fit.

Abstract

Overall, the abstract effectively summarizes the work. A few clarifying points are requested.

Line 34: the covalent-closed-nature of a putative circular RNA cannot be inferred from RNA sequencing alone (including FLDS), neither can the circularity of a putatively circular RNA. The features used for inferring circularity are confounded by concatemeric/repetitive sequences, so the two types of species cannot be distinguished by sequencing. For these reasons, I believe that the “covalently closed circular” phrase needs to be couched in less declarative tones e.g. “apparently covalently closed circular” throughout the manuscript when referring to RNA elements from previous studies, including for the Obelisk paper (which are described as cccRNAs in Line 35 too) - where neither the apparent covalent or circular molecular features of these elements were tested appropriately.

We use “apparently” at Lines 34, 37, and 52 to emphasize that covalent circularity cannot be established from sequencing data alone and to align our terminology with the reviewer’s suggestion.

Line 41: “Ch” typo

We thank the reviewer for noting this typo. The error has been corrected.

Line 44: If HsObs HsOblins don’t share “significant sequence similarity” with Oblin-1 then how were these elements recognized as Obelisks (implied in Line 42). I think the authors should a) explain how these elements were identified, and b) give a metric to what they mean by “no significant sequence similarity”

We revised the relevant sentences to clarify how HsObs were identified as Obelisks. The text now states: “Despite lacking sequence similarity to known Oblins, HsObs shared hallmark features of the Obelisk family, including ~1 kb genome size, extensive rod-like RNA secondary structure, and the predicted fold of the encoded protein, HsOblin.”

Line 44: in my opinion, “significant” is a sacrosanct word in science that implies statistical hypothesis testing, if such as test was conducted then please state which test was used and what p-value cutoff was used, otherwise, please use a different word to describe the nature of the sequence similarity.

This point has been addressed as noted above.

Line 44: which sequence, nt or aa?

This point has been addressed as noted above.

Line 45: “Metatranscriptomes” are being queried rather than “Metagenomes”

We thank the reviewer for pointing this out. Indeed, the original text incorrectly referred to “metagenomes”; We have corrected this to “metatranscriptomes” in the revised manuscript.

Line 46: “Substantially” what approximate fold change

As suggested, we added the approximate fold change (‘about two-fold’) in the abstract (L47 in the revised manuscript).

Line 48: If the authors need more words in the abstract, I think the line pertaining to the Yellowstone MTX could be instead a clause added to the “substantial expansion” sentence

As suggested, we revised the abstract to improve conciseness and flow. The reference to the Yellowstone metatranscriptome was removed, as it was not essential to the main message and helped reduce the overall word count.

Introduction

Overall, the introduction is written well and gives a good overview of the field to the reader. Several citations are missing throughout (which have been cited previously but should be repeated as specific points are made). My major concerns with the introduction are 1) the use of “cccRNA” to describe entities of un-validated circularity of covalent structure (such as Obelisks), and 2) the breadth of the introduction. For this second point, which I guess is mostly editorial, I defer up to the authors for how they wish to structure their manuscript, but I think the introduction could be shortened for brevity by making it less of a “review” of the field.

We appreciate the thoughtful comments. With regard to the usage of cccRNA, we now indicate that the discovered replicons are ‘apparently’ cccRNAs and added a separate paragraph toward the end of the Introduction explaining and emphasizing the caveats.

With regard to the breadth of the Introduction, we appreciate the comment and have reviewed the text for possible cuts. In our opinion, very little could be omitted without leaving out what we consider useful background information. Therefore, considering that the Introduction is not long in the absolute sense and the entire paper is concise, we felt that it was useful to present a more or less broad landscape of this burgeoning field (of which the reviewer is surely well aware, having made one of the pivotal contributions). Thus, no cuts, given that the reviewer marks this suggestion as discretionary.

First paragraph: this is a nice introduction to viroids, though perhaps is too much background on viroids vs the subject matter of the manuscript. E.g. the details on viroid genome processing could be omitted for brevity

Second paragraph: this is also good, and could also likely be shortened for brevity due to relevance to the manuscript. Please also re-add the citations present in the first paragraph for the specific examples described in this second paragraph (see below).

Line 66: “bona fide” should be in italics

Fixed.

Line 67: add “virusoid” citation

Included.

Line 69: add “retroviroid” citation

Included.

Line 72: add “retrozyme” citation

Included.

Line 82: add recent HDV-like citations

Included.

Third paragraph: again, a nice intro into HDV but could also be condensed down

Line 90/91: add citation for CRISPR x cccRNA and an approximate count of what “several”

means

Citation added, We indicate in the revision that 9 distinct groups were targeted by CRISPR spacers (L89-90 in the revised manuscript).

Line 94: add citations for HDV-like elements of which there are several

Included.

Line 95: add citations for ambiviruses

Those were already in the original text, in the next line.

Fourth paragraph: again, this is a nice introduction/summary of ambiviruses but I'm not sure all this detail/context is needed for this manuscript

As per the above, we selected to keep it. The details are really quite limited.

Main Text

Overall, the main text is written well and is easily understood. Editorially, it is clear that the two first authors each wrote different parts of this section - that is completely fine and expected - but it does mean that the first half doesn't reference the later "methods" section and the latter half does; which makes my feedback over this section inconsistent. Having next read the "methods" section, some of the points I make here are addressed and could be referenced by stating "see methods" as the authors see fit. Scientifically, everything is well written and clear. I am particularly interested in the finding that the HsObelisks appear to not be detectable by the original Oblin-1 pHMM (though this is not explicitly stated in this section - please make this clearer if this is indeed the case) - which is very cool! Additionally, recent efforts (Logan bioRxiv, Chikhi et al. 2025) have built a new Oblin-1 pHMM and Obelisk database, it would be interesting to see how these compare to your findings but I also do not think this is a required comparison to make, please at least add a citation to this work reflecting the evolving nature of this space. You should probably also cite Edgar et al. 2022 when referencing HDV-like MTX mining.

We appreciate this constructive comment. We retrieved the HMM profile for Oblin-1 from the Logan depository (all Oblin-1 HMMs deposited in s3://logan-pub/paper/Obelisk/12_Obelisk_Logan_Run2) and ran hmmsearch against our and

previously reported Oblin-1 proteins. Oblin-1 proteins from most clades including with known members were retrieved with high confidence. However, the ‘logan’ profiles only recovered a small fraction of the newly discovered clades, and only with low confidence. This result is comparable to the results of our initial search where we performed structure prediction for low confidence hits followed by construction of clade specific profiles. The clade harboring the HsOblin-1 is an exception as about 70% of it’s members were detected with the logan HMM profiles albeit at moderate to high e-values. These observations highlight the advantage of our structure-guided approach to for the validation of low-confidence matches and the construction of clade-specific HMMs. We have added this observation to the Discussion and to Fig. S14.

Both the Logan preprint and Edgar et al are cited in the revision.

General point: where possible/known, please list the version numbers of the softwares used

Added where applicable.

General point: where possible/known, please list the dates of accession to databases (e.g. BLAST) when they were used

Added where applicable.

Line 120: same comment as before, we cannot determine “circularity” from these data

This point has been addressed as noted above.

Line 121: RCR is not the only conceivable mechanism by which a cccRNA might replicate. Could be discontinuous RCR (Jain et al. 2020) or rolling hairpin replication (Tattersal et al. 1976), or other mechanisms. Instead of diving into the weeds of putative replicative mechanisms, I would just explain that RNA-dependent RNA replication requires the generation of a template-substrate dsRNA duplex (regardless of replicative mechanism), which can be captured by dsRNA sequencing

Thank you for the suggestion. We revised the text accordingly to avoid implying a specific replication mechanism and now state that RNA-dependent RNA replication, regardless of the precise mechanism involved, requires the formation of a template–substrate dsRNA duplex, which can be captured by dsRNA sequencing.

Line 128: add citations to previous methods that have used terminal repeats for the same purpose

Because ccfind does not have a formal citation, we added a reference directing readers to the “Code availability” section.

Line 129: BLAST doesn't do significance tests so I would avoid using the word “significant” and instead use “substantial” (or equivalent) and quote an E-value/BitScore/etc threshold used for this conclusion

Line 131: what units is the “<10” value referring to? Likewise, what units are in the “10x” value?

We thank the reviewer for pointing this out. In the revised manuscript, we clarified that “<10” and “10x” refer to the average read depth (reads per nucleotide), and we updated the corresponding text accordingly.

Line 132: which RNA folding program was used for this prediction and what settings (e.g. if viennaRNA was used, was “circularity” enforced?).

Added (L707-714 in the revised manuscript).

Line 133: this sounds like an interesting analysis, please explain it a little more for the reader and also cite the conclusions from your measurement of the “evenness of mapping” (including units) and any relevant citations too

We thank the reviewer for this helpful comment. In the revised manuscript, we added a brief explanation of how mapping evenness was evaluated, including the normalized coefficient of variation (CV) and entropy (both dimensionless measures) thresholds used for contig selection, and we stated the conclusions drawn from these metrics. Specifically, we explain that relatively uniform read coverage is used as a selection criterion based on our empirical observation that, in FLDS data, contigs derived from *bona fide* dsRNA molecules tend to show more even coverage than those originating from non-dsRNA sources. This behavior has been documented in previous FLDS studies, which we now cite. We also refer the reader to the Methods section for additional details.

Line 138: this method cannot differentiate between circles and concatemers.

We revised the phrasing at Line 138 by replacing “circular RNA replicon” with “cccRNA replicon,” consistent with our earlier clarification that covalent circularity has not been experimentally demonstrated and that FLDS cannot distinguish circles from concatemers.

Line 167:

which covariance models / ribozymes were searched - please add Rfam accessions or and/add a supplementary file

We have added the Rfam file to the supplementary dataset at zenodo (infernald/ rfam_cm) and pointed to the file in the Methods section.

Line 172/173: please add AF2 and AF3 settings and citations. For AF2 please also add how the MSA was constructed (if it was) for structural inference

Both AF2 and AF3 settings are given in the Methods section and comprise mainly default settings. To keep the main text simple, we have only added the citations to the main text (L193 in the revised manuscript).

Line 177: how was the sequence alignment (aa or nt) constructed, and what sequence was this ORF aligned to? Was an alignment against the original Oblin-1 pHMM attempted?

We thank the reviewer for pointing this out. The structural alignment of HsOblin-1-Oi and Oblin-1 protein structural models was performed by the MatchMaker tool in ChimeraX program. We added the following clarification in the method, and a (See Methods) reference to the main text.

“The MatchMaker tool of ChimeraX was used to align and compare the predicted structures of HsOblin-1-Oi and Obelisk Oblin-1 proteins. The Oblin-1 model was aligned to the confidently predicted alpha helical regions aa51-148 of HsOblin-1-Oi. The RMSD value for all aligned pairs was 3.817 Å. For comparison, the “best” Foldseek hit A0A481Z095 (E-value 0.31) showed the RMSD of 12.670 Å when aligned to the HsOblin-1-Oi aa51-148 model by the MatchMaker tool.”

Line 140: “circular”

Line 145: “significant”

Line 147: coverage units

Line 171: “significant”

Line 183: “significant”

We have addressed these editorial points and revised the wording as explained above.

Line 196: please add how different the pLDDT scores were +/- RNA

Line 202: likewise, please add pLDDT/pAE numbers to this text

The confidence score values are added to the revised main text.

Line 215: please state which tool / settings / database was used for the spacer search

We thank the reviewer for this comment. We have added a dedicated section in the Methods specifying the tools, databases, and search parameters used for the CRISPR spacer search. During the revision, we also noticed and corrected an error in our original description. Specifically, we had stated that 919 spacers were obtained from five samples; however, the correct numbers are 720 spacers obtained from four samples. We apologize for this mistake.

Line 254: how is coverage calculated here?

The coverage here is the fraction of a monomeric Obelisk covered by the Oblin-1 ORF. We adapted the text accordingly.

Line 280: how does the RNA secondary structure map to the Oblin-1 ORF, is it similar to other Obelisks where Oblin-1 is predicted to be largely self-complimentary?

This is an interesting question. We analyzed which fraction of the paired bases for which at least one resides within the Oblin-1 ORF which has a counterpart which resides as well in the Oblin-1 ORF. On average, 82% of the paired ‘Oblin-1 ORF bases’ are self-complementary. We included this information within the text and as supplementary figure (Fig. S15).

Line 291: the value of 51 nt from the 5` end is eye-catching because that is the arbitrary indexing convention of the Oblin-1 A(51)UG of the sense strand from the 5` end of the Obelisk genome (50nt upstream sequence, then start of Oblin-1 at 51nt) - please add a note to specify if this median distance is artifactual from this arbitrary convention and/or please clarify if these 51nt are from

the STOP of the Oblin-1 in the antisense strand (the language of 5` +/- sense/antisense gets confusing)

This could be an interesting link. But given the variation in the observed distance (Fig. S7 B, varying between several hundred to 1 nt distance), we strongly believe that this median number of 51 is artifactual. We added in the text that the median distance refers to the start codon position of the Oblin-1 ORF. Given that we don't refer to the arbitrary indexing convention of Oblin-1, we assume that including it might distract the reader rather than being helpful.

Line 293: what frequencies?

We have added the frequencies of ribozymes embedded within the Oblin-1 ORF in the text.

Line 320: "significant" and do the other alpha-helical non-Ob-1 ORFs contain leucine zipper motifs?

'significant': changed to 'detectable sequence similarity'

A hmmer search against pfam indicated no classical leucine zipper hits. We ran in addition coiled-coil predictions for non-Oblin-1 proteins and inspected them for leucine stretches across heptads. We did not observe many putative leucine zippers when applying the following criteria: at least three heptad repeats for which a leucine is observed at the 'a' position and leucine/isoleucine at the 'd' position, in at least 50% of the 'a' and 'd' positions and at least three matches either at the 'a' or the 'd' position. A variation in 'a' and 'd' position is also visible in the reviewers original publication (Fig. 2C), indicating that the observed leucine zipper of Oblin-2 is more relaxed than the classical zipper. We added this observation to the Results, and the coiled-coil and leucine zipper results are provided in the Zenodo repository as supplementary data.

Discussion

The discussion is well written and aptly summarizes the work without making unsubstantiated claims. I suggest a few edits to qualify some statements.

Line 354: I wouldn't say finding the spacer has "clinched" that that cccRNA has a bacterial host because we still don't really know how well cccRNAs are surveilled by the CRISPR system (or

really much about RNA-surveilling spacer acquisition), or have a notion for how reliable spacer-based-host-ID really works, so I would change the language here to reflect that this observation supports the putative host being bacterial.

'Clinching' changed to 'strongly suggesting'.

Line 364: I think it's a little premature to conclude from the hallucinations of AF3 that there's a specific HsOb RNA binding interaction. I would further qualify this sentence to reflect how very preliminary this association is.

We thank the reviewer for this comment. We agree that functional conclusions regarding RNA binding based on structural prediction alone would be premature. In response, we revised the Discussion to remove AlphaFold3-based functional interpretations and to substantially tone down statements regarding RNA binding. We also revised the Supplementary Figure S1 title to avoid implying RNA-binding activity and to limit interpretation to changes in model confidence metrics. We believe that this revision appropriately addresses the reviewer's concern by clearly distinguishing structural compatibility from experimentally validated function and by avoiding overinterpretation of predictive models.

Line 377: I think the authors should explain this Hydrogenobaculaceae association more clearly (that the MTX suggests this is the only -?- genus of bacteria in the sample) either here or in the main text

We thank the reviewer for this helpful suggestion. In the revised manuscript (L441-449), we explicitly state the basis for the association with Hydrogenobaculaceae by adding a brief explanation of the SSU rRNA profiling results, which show that Hydrogenobaculum overwhelmingly dominates the microbial community in H5 and is the majority lineage in Oi.

Methods

The bioinformatic methods sections are written well with appropriate details and a lot of the points raised in the earlier main text section are answered here. The experimental methods sections however lack critical detail on almost every point. The authors need to completely re-write the experimental methods sections such that every single step of the protocols carried out can be

replicated by another lab. Note, it is also insufficient to cite previous work for methods details.

Line 389: what cultivar? Which leaves? Where was the sample taken from in the leaf and how?

Regarding the plant materials and sampling methods, we have added the following details to the "Method" section (Line 462-466 in the revised manuscript):

"Chrysanthemum morifolium plants were grown from seeds of the commercial line 'Nihon Kogiku' (mixture) obtained from Sakata Seed Corporation (Yokohama, Japan) and maintained in a greenhouse at the National Agriculture and Food Research Organization (NARO). Upper fully expanded leaves infected with Chrysanthemum stunt viroid (CSVd) were collected. "

Line 390: what volume of LN2? what method was used for nucleic acid extraction? Buffers, volumes, incubation temperatures, citations, etc..

Line 391: what cellulose-based method was used for dsRNA enrichment? Buffers, volumes, incubation temperatures, citations, etc..

Line 392: where were these enzymes sourced from? What volumes, temperatures, buffers etc. Were the enzymes used simultaneously or sequentially? How were the enzymes removed from the reaction mix prior to library prep? What are the expected nucleic acid depletions from using these enzymes?

Line 392: what settings were used for fragmentation? Was the sample cooled? What volume was used? Etc etc etc

Line 393: what is the U2 primer? Is it cited in the supplemental tables? How was it ligated? How was the material purified etc etc.

Line 394: which kit was used? which enzyme was used, what experimental settings, how long, how was it purified?

Line 395: what PCR cycling conditions were used, what enzyme, what primers, how was indexing done, which indices were used?

Line 396: please provide every detail of the library prep

Line 397: which Novaseq was used, how many reads, was PhiX used as spike-in, what was the read length, what was the sequencing strategy (SE or PE), what was the name of the company?

We thank the reviewer for requesting additional details regarding sequencing. We have now specified the sequencing platform, read configuration, PhiX spike-in ratio, and the number of reads obtained per library in the Supplementary Methods. These additions further clarify

the sequencing strategy and data yield for both FLDS version 2 and version 3.

Line 403: please summarize and then provide these “geochemical and community structure” data - and explain how they were determined.

We thank the reviewer for this helpful suggestion. In the revised manuscript, we added a concise description of how the geochemical parameters and microbial community structures of the hot spring samples were obtained and summarized these data in a new supplementary table (Table S3).

Line 404: this line appears to be irrelevant to this particular manuscript?

We thank the reviewer for pointing this out. Indeed, the sentence at Line 404 was not directly relevant to the analyses presented in this manuscript and has been removed in the revised version.

Line 411: please provide a link to ccfind

We added the requested link to ccfind in the Methods section (L497 in the revised manuscript).

Line 413: default BLASTn/x settings?

We added a note to the Methods indicating that, apart from the parameters explicitly stated in the text, BLASTN and BLASTX searches were performed using default settings.

Line 459: I think there's a typo here, what does “4.8 putative proteins” refer to?

‘million’ added. ‘...about 4.8 million putative proteins (at least 60 aa long) from 8.9 million recently published putative cccRNAs...’

Line 600: There's a typo throughout the manuscript referring to “Jupiter” plots as “Jupyter” plots - the name refers to the banded structure of the gas giant, not the notebook software.

Corrected. Thanks for pointing out.

Line 636: thank you for making your data publicly available.

Zenodo link is now included.

Figures

The figures are clear and their captions make sense. I would request that the figures are rendered in a higher DPI and time permitting, with larger components and better color contrast for folks hard of sight.

We thank the reviewer for this suggestion. The figures in the revised manuscript are shown at reduced resolution due to file size limitations during submission. All figures will be supplied at high DPI in the final version, in accordance with the journal's guidelines, and we will consider adjustments to layout and color contrast to improve readability where appropriate.

Reviewer #2 (Remarks to the Author):

In this work, the authors explored dsRNA metatranscriptomes constructed by Fragmented and primer- Ligated DsRNA Sequencing (FLDS) , a method for selectively sequencing of dsRNA, a typical replicative intermediates of RNA replicons. Your main text is so difficult to reads a need a significant improvement. It is almost impossible to follow your story.

Did you tried to amplify your dataset with other samples ? s not visible from how many samples you started your research

We thank the reviewer for this comment. To improve clarity and readability, we revised the Results section by reorganizing subsection titles and rewriting the opening sentences to better guide the reader through the initial validation and subsequent analyses. We also clarified the scope of the dataset by explicitly stating the number of hot spring samples analyzed (11 samples) at the beginning of the relevant subsection. We hope that these revisions improve the narrative flow and make the study easier to follow.

Reviewer #3 (Remarks to the Author):

Uruyama and colleagues here present a significant addition to the small but growing number of studies that have identified cccRNAs known as Obelisks. Their work relied on a unique strategy of looking within samples from acidic hot springs and used metatranscriptomes generated by Fragmented and primer-Ligated dsRNA Sequencing (FLDS).

Their results include a novel Obelisk found in the hot springs metatranscriptomes and were further expanded upon by searching for related cccRNAs in a large collection of such sequences. They also identified and characterized several families of novel Obelisk-associated proteins.

These findings should be useful for guiding further identification and characterization of Obelisks and their sequence features, including encoded proteins.

We thank the reviewer for the positive and encouraging evaluation of our work. We are pleased that the importance of identifying a new Obelisk from acidic hot spring dsRNA metatranscriptomes and expanding the analysis to related cccRNAs was recognized. We also appreciate the reviewer's acknowledgment of our characterization of new Obelisk-associated protein families. We believe these findings indeed will help facilitate further discovery and functional studies of Obelisks, and we are grateful for the reviewer's supportive comments.

Their manuscript may benefit from considering the following:

-It is not obvious why they specifically chose to look within hot spring environments.

We thank the reviewer for this helpful comment. To clarify our rationale for focusing on acidic geothermal springs, we added text to the Introduction (L117-124 in the revised manuscript) explaining that these environments are known to support replicating RNA elements with linear genomes, indicating that RNA replicators can persist there. We also explain that cccRNAs are typically difficult to detect by homology-based searches and that FLDS, which relies on structural signatures, is particularly suitable for identifying such elements. We believe these additions make our motivation for choosing hot spring samples clear.

-The significance of predicted structural differences at 37c and 80c could be explored or discussed further.

We added a brief note in the Results (L183-186 in the revised manuscript) to clarify that the 37°C and 80°C RNAfold predictions are interpreted qualitatively, indicating reduced base pairing but retention of the overall rod-like topology at high temperature.

-The significance of potential RNA-binding regions of HsOblin-1 could be explored or discussed further.

We agree that the RNA-binding potential of HsOblin-1 requires cautious interpretation. In the revised manuscript, we revised and streamlined the Discussion to avoid overinterpretation of structural predictions and to explicitly state that there is currently no direct experimental evidence supporting RNA-binding activity of HsOblin-1 (L401–408). We also revised the title of Supplementary Figure S1 accordingly. Any proposed RNA-binding role of HsOblin-1 is therefore framed as hypothetical, and no functional claims are made.

-The lack of CRISPR spacer matching is somewhat surprising given the number of Obelisk sequences that were included in their search, however previous studies have also struggled to find matches. Perhaps more info could be provided on the spacer database searched, and perhaps trying other available databases (eg that from JGI).

We appreciate this valuable suggestion. We searched, in addition to the smaller curated spacer databases, the current JGI high quality spacer set and found a number of highly confident matching spacers. We have incorporated these results in the main text and several supplementary figures (Fig. S11-S13), showing spacer characteristics (e.g. number of mismatches for spacer/Obelisk alignments (if any), associated CRISPR types and hosts), location of spacer matches on the Obelisks and Obelisks with spacers matched to the Oblin-1 phylogeny. Interestingly, we got many spacer matches against previously published Obelisks, consistent with the fast-expanding JGI/IMG spacer database. We thank the reviewer again for this constructive suggestion.

-The lack of homologous matches for the non-Oblin-1 proteins despite searching several databases via both profile and structure based comparisons is a bit perplexing, but may just represent the unique novelty of these putative proteins.

The lack of homologs for these predicted proteins may not be too surprising. This is a

common situation for small alpha-helical domains. Yet, we are convinced that some of these are real, given their conservation within specific Obelisk lineages. We expanded the discussion of this point.

-In the discussion they comment on the likelihood of potential RNA binding sites and the Oblin-1 protein being involved in Obelisk replication, but their reported work did not directly address replication.

We agree with the reviewer that our data do not directly address replication. We revised the Discussion to clarify that the potential involvement of Oblin-1 in replication and RNA binding is speculative and based only on predicted structural features, and that no replication mechanism is demonstrated in this study.